# Regulatory T cells infiltrate the tumor-induced tertiary lymphoïd structures and are associated with poor clinical outcome in NSCLC

Priyanka Devi-Marulkar[1,2,3,14], Solène Fastenackels[4,5,6], Pierre Karapentiantz[1,2,15], Jérémy Goc [1,2,3,16], Claire Germain[1,2,3,17], Hélène Kaplon[1,2,3,18], Samantha Knockaert[1,2,3,18], Daniel Olive[7,8], Marylou Panouillot[4,5,6], Pierre Validire[3,9], Diane Damotte[1,2,3,10], Marco Alifano[3,11], Juliette Murris [1,2,12,13], Sandrine Katsahian[1,2,12,13], Myriam Lawand[1,2,3] & Marie-Caroline Dieu-Nosjean [1,2,3,4,5,6✉]

On one hand, regulatory T cells (Tregs) play an immunosuppressive activity in most solid tumors but not all. On the other hand, the organization of tumor-infiltrating immune cells into tertiary lymphoid structures (TLS) is associated with long-term survival in most cancers. Here, we investigated the role of Tregs in the context of Non-Small Cell Lung Cancer (NSCLC)-associated TLS. We observed that Tregs show a similar immune profile in TLS and non-TLS areas. Autologous tumor-infiltrating Tregs inhibit the proliferation and cytokine secretion of CD4$^+$ conventional T cells, a capacity which is recovered by antibodies against Cytotoxic T-Lymphocyte-Associated protein-4 (CTLA-4) and Glucocorticoid-Induced TNFR-Related protein (GITR) but not against other immune checkpoint (ICP) molecules. Tregs in the whole tumor, including in TLS, are associated with a poor outcome of NSCLC patients, and combination with TLS-dendritic cells (DCs) and CD8$^+$ T cells allows higher overall survival discrimination. Thus, Targeting Tregs especially in TLS may represent a major challenge in order to boost anti-tumor immune responses initiated in TLS.

[1] Sorbonne Université, UMRS 1138, Cordeliers Research Center, Paris, France. [2] Université de Paris, UMRS 1138, Cordeliers Research Center, Paris, France. [3] Laboratory "Cancer, Immune Control, and Escape", Inserm U1138, Cordeliers Research Center, Paris, France. [4] UMRS1135 Sorbonne Université, Faculté de Médecine Sorbonne Université, Paris, France. [5] INSERM U1135, Paris, France. [6] Laboratory "Immune Microenvironment and Immunotherapy", Centre d'Immunologie et des Maladies Infectieuses (CIMI-Paris), Paris, France. [7] Inserm U1068, CNRS, UMR7258, Institut Paoli-Calmettes, Aix-Marseille University, Marseille, France. [8] Laboratory « Immunity and Cancer », Centre de Recherche en Cancérologie de Marseille (CRCM), Marseille, France. [9] Department of Pathology, Institut Mutualiste Montsouris, Paris, France. [10] Department of Pathology, Assistance Publique-Hôpitaux de Paris (AP-HP), Cochin hospital, Paris, France. [11] Department of Thoracic Surgery, Assistance Publique-Hôpitaux de Paris (AP-HP), Paris, France. [12] HeKA, INRIA, Paris, France. [13] Hôpital Européen Georges-Pompidou, Unité d'Epidémiologie et de Recherche Clinique, Assistance Publique-Hôpitaux de Paris (AP-HP), Inserm, Centre d'Investigation Clinique 1418, Module Epidémiologie Clinique, Paris, France. [14] Present address: Institut Curie, Paris, France. [15] Present address: Inserm, Sorbonne Université, université Paris 13, Laboratoire d'informatique médicale et d'ingénierie des connaissances en e-santé, LIMICS, F-75006 Paris, France. [16] Present address: Joan and Sanford I. Weill Department of Medicine, Division of Gastroenterology and Hepatology, Department of Microbiology and Immunology and The Jill Roberts Institute for Research in Inflammatory Bowel Disease, Weill Cornell Medicine, Cornell University, New York, USA. [17] Present address: Biomunex Pharmaceuticals, Paris, France. [18] Present address: Translational Medicine Department, Institut de Recherches Internationales Servier, Suresnes, France. ✉email: marie-caroline.dieu-nosjean@inserm.fr

Tumors are sustained by a complex network of interactions between tumor, stromal, and immune cells. The immune system is able to detect the tumor cells[1], and plays an important role in tumor rejection. The composition and density of intra-tumoral immune cells are highly heterogeneous and have an influence on the disease outcome of cancer patients[2,3]. Not only the infiltration, but also the organization of cells in the tumor microenvironment into tertiary lymphoid structures (TLS) is a major phenomenon for the long-term survival of NSCLC patients[4–6]. Indeed, tumor-associated TLS represent a privileged site for T cell differentiation and activation. Furthermore, a high density of TLS has been shown to be associated with a T helper 1 (Th1) and cytotoxic immune signature in lung[7], breast[8], and gastric cancers[9], indicating that TLS may imprint the local immune microenvironment.

To escape immune responses, tumors can develop several mechanisms of regulation via, for instance, the recruitment of regulatory T cells (Tregs)[10]. In human tumors, Tregs are recruited via CCL17/CCL22 and CCR4 interaction[11–13]. Tregs suppress dendritic cell (DC) and effector T cell functions via the production of cytotoxic molecules (granzymes A and B, perforin), the secretion of immunosuppressive cytokines (i.e., IL10 and TGF-β), and/or the expression of immunoregulatory receptors (i.e., CTLA-4, LAG-3)[14]. However, the prognostic value of Tregs in cancer patients is a matter of debate[15]. It is highly influenced by the phenotype, histological parameters, areas in the tumor[16], and the cancer type[17]. Moreover, the ratio of cytotoxic T cells to Tregs is considered as a stronger prognostic parameter than Tregs alone[18]. Considerable research has been carried over the past few decades in understanding the molecular basis underlying the immune regulation by Tregs. Mostly, the primary mechanism employed by Tregs likely depends on the disease settings, the target cell type, the local inflammatory environment, and anatomical location. It has also been observed that Tregs undergo a paired differentiation with helper T cells they suppress, by co-opting the transcriptional program to undergo functional specialization in the periphery[19]. Furthermore, the presence of Tregs has been found to influence the formation of the TLS in tumors[20]. Thus, it becomes critical to understand the qualitative and quantitative role of Tregs in the shaping of the anti-tumor immune response.

In this study, we observed the presence of Tregs in the stroma, including TLS of lung tumors. In contrast to blood and to a lesser extent to lymph nodes (LN) Tregs, most tumor-infiltrating Tregs (TIL-Tregs) displayed an activated and effector-memory phenotype without major differences in TLS and non-TLS areas. TIL-Tregs are functional since they inhibit the proliferation and cytokine production of autologous CD4$^+$ conventional T cells (Tconv). High Treg infiltrate is associated with a poor clinical outcome of NSCLC patients. A deeper analysis of Treg localization also showed that a high density of TLS-Tregs is correlated with a short term-survival of NSCLC patients. When combined with TLS mature DC or CD8$^+$ T cells, we observed that all three immune markers allow the identification of patients with the highest risk of death. Altogether, these data suggest that Tregs are capable of local immunosuppression in the tumor microenvironment, especially in TLS where anti-tumor immune responses are initiated.

FoxP3$^-$ T cells in Fig. 1b; CD8$^+$ T cells in Fig. 1e, serial sections), to a lesser extent in TLS (CD3$^+$ FoxP3$^-$ T cells in Fig. 1c–f, serial sections), and rarely in tumor nests (CD3$^+$ FoxP3$^-$ T cells in Fig. 1a; CD8$^+$ T cells in Fig. 1d, serial sections). The distribution of Tregs was similar to Tconv, i.e., massively in the stroma outside TLS (CD3$^+$ FoxP3$^+$ cells Fig. 1b), to a lesser extent in TLS (Fig. 1c), and rarely in tumor nests (Fig. 1a). TLS-Tregs also expressed CD62L, the TLS-homing molecule[22] as shown by multiplex IF (Fig. 1g) and flow cytometry analyses (CD3$^+$ CD4$^+$ FoxP3$^{high}$ CD25$^{high}$ CD62L$^+$ cells, Fig. 2b). We confirmed that TLS-Tregs also express lymphoid chemokine receptors (CCR7, CXCR5) as well as Tfh markers such as PD-1 and ICOS (Fig. 2d).

The ratio non-TLS /TLS was in favor of non-TLS for both Tregs and CD4$^+$ Tconv, and even more pronounced for CD4$^+$ Tconv (Fig. 2a), indicating that most of them preferentially homed in non-TLS areas.

From fresh lung tumor samples, we confirmed that CD3$^+$ FoxP3$^{high}$ CD25$^{high}$ cells were CD127$^-$, massively CD4$^+$ (not CD8$^+$) (Fig. 2c), in accordance with the phenotype of human natural CD4$^+$ Tregs[23–25]. Finally, cells were CD45RA$^-$ and most of them expressed a high level of the HLA-DR molecule indicating that TIL-Tregs are mainly activated with a memory phenotype.

These data show that, Tregs migrate in every subarea of the lung tumor but in a different proportion, similarly to CD4$^+$ Tconv.

**Distinct stages of Treg differentiation relative to their anatomical sites in NSCLC.** The percentage of FoxP3$^+$ Tregs among total CD4$^+$ T cells is strongly heterogeneous and higher in lung tumors (median = 14%) compared with those at distant sites, i.e., non-tumor distant lung (NTDL, median = 5%), lymph nodes (LN, median = 8%) and blood (median = 7%) samples from NSCLC patients (Fig. 3a), suggesting an active recruitment and/or local proliferation of Tregs within the tumor. Tregs are mainly of effector-memory and central-memory phenotype in all anatomical sites based on the differential expression of CCR7, CD45RA, CD27, and CD28 (Supplementary Fig. 3a). However, the ratio is reverted in favor of effector-memory stage in lungs (i.e., tumor and non-tumor) compared with distant sites (blood and LN, Fig. 3b).

We then compared the differentiation stage of Tregs and CD4$^+$ Tconv relative to TLS (CD62L$^+$ cells in TLS, and CD62L$^-$ cells in non-TLS areas). In non-TLS areas, Tregs were mainly of EM1 phenotype, and to a lesser extent, CM and EM4 (earliest EM stage), whereas these three T cell differentiation stages were equally distributed on CD4$^+$ Tconv (Fig. 3c, upper panel). Of note, terminally differentiated TEMRA cells (PE1/2 and E stages) were not detected for any of the CD4$^+$ T cell populations in lung tumors. The Treg profile in TLS was similar to those in non-TLS with mainly EM1 and CM stages. This observation contrasts to CD4$^+$ Tconv, which were present in an earlier stage of differentiation, i.e., with more naïve and CM, and less EM1 and EM4 in TLS than that on non-TLS (Fig. 3c, lower panel).

Taken together, Tregs are mostly at the memory stage of differentiation in tumors and distant non-tumor sites. However, in contrast to CD4$^+$ Tconv, the profile of Tregs is quite unchanged in TLS and non-TLS areas, suggesting similar effector functions in each sub-areas of the tumor.

**Functional orientation of TIL-Tregs is remarkably different from their counterparts in blood but not in NTDL.** With a prevalent infiltration of Tregs with an EM phenotype in tumors, we aimed to determine whether they may exert a putative effector function through specific mechanism(s) in tumors compared with

## Results
**Tregs infiltrate distinct tumor areas of NSCLC patients.** We first evaluated the presence, localization, and frequency of Tregs[21] relative to other tumor-infiltrating T lymphocytes in NSCLC. First, the vast majority of non-Tregs (named conventional T cells, Tconv) were observed in the tumor stroma (CD3$^+$

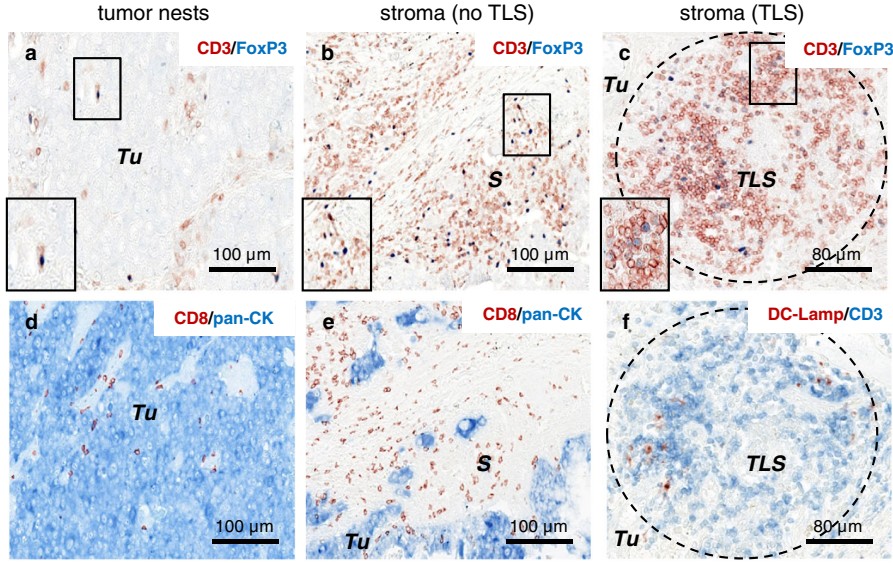

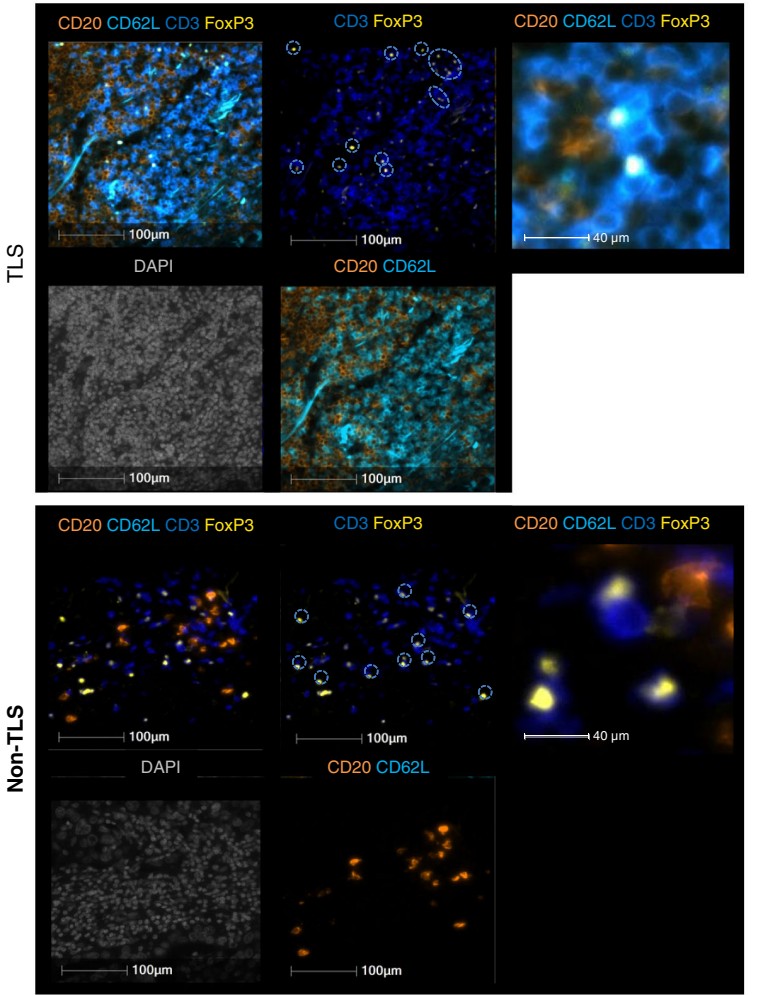

**Fig. 1 Tregs infiltrate different lung tumor areas.** Double IHC (**a–f**) and 5-plex IF staining (**g**) on paraffin-embedded lung tumor sections. Presence of CD3+ (red) FoxP3+ (blue) T cells in tumor nests (**a**), stroma (**b**), and TLS (**c**). Presence of CD8+ T cells (red) in pan-cytokeratins+ (blue) tumor beds (**d**) and pan-cytokeratins- stroma (**e**). **f** Detection of DC-Lamp+ (red) mature DC and CD3+ (blue) T-cell rich areas of TLS (encircled in black dotted line). Serial sections: **a–d**, **b–e**, and **c–f**. **g** 5plex-IF staining (DAPI in gray, CD3 in dark blue, FoxP3 in yellow, CD20 in orange, and CD62L in light blue) in TLS and non-TLS areas of NSCLC section. Dotted circle shows CD3+ FoxP3+ Tregs in TLS and non-TLS areas. S stroma, TLS tertiary lymphoid structure, Tu tumor nest.

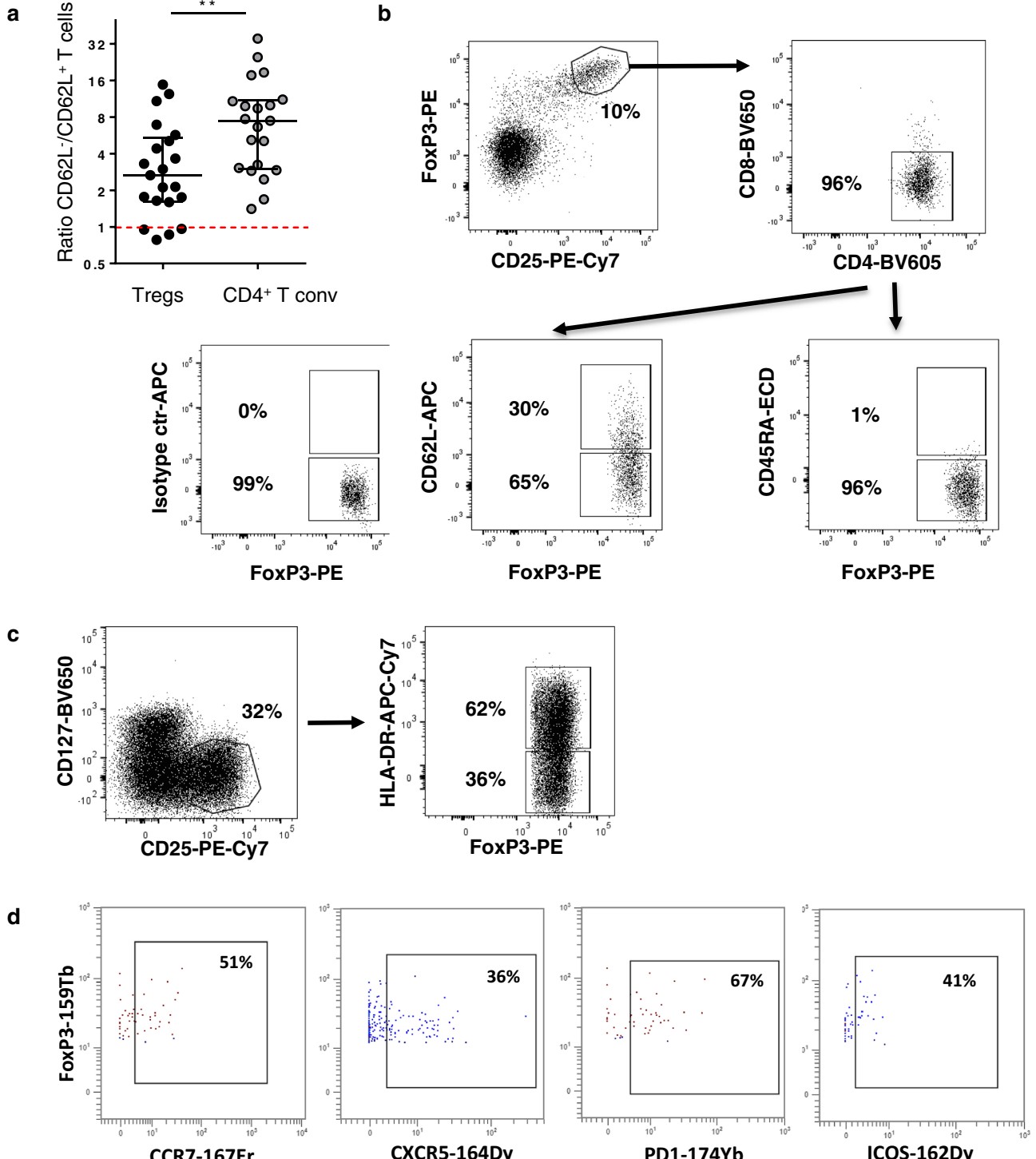

**Fig. 2 Phenotype of Tregs in NSCLC. a** Ratio of non-TLS to TLS CD3+CD25^bright^FoxP3^bright^ Tregs and CD4+ Tconv was determined according to the differential expression of CD62L ($n = 21$ patients). Data are represented by median with interquartile range. Wilcoxon–Mann–Whitney $U$ test; **$P = 0.0079$. **b**, **c** Characterization of CD3+CD25^bright^FoxP3^bright^ cells according to CD4, CD8, CD62L, CD45RA, CD127, HLA-DR, and CD45ra expression by flow cytometry on fresh NSCLC tumors. **d** Illustration of the expression of CCR7, CXCR5, PD-1, and ICOS on CD3+ CD4+ CD8− FoxP3^bright^ CD25^bright^ CD127- CD62L+ TLS-Tregs.

other anatomical sites. Therefore, the expression of 120 genes was compared in Tregs infiltrating tumors, NTDL, LN, and blood samples from NSCLC patients. The most important number of genes differentially expressed was observed on Tregs isolated from tumors versus blood samples (Fig. 4a, Supplementary Fig. 4). Most of them were overexpressed in tumor-infiltrating

Tregs, and included co-stimulatory and inhibitory molecules (e.g., *4-1BB, OX-40, BTLA, PD1, GITR, PD-L1, Tim-3, B7-H3,* and *LAG-3*), chemotactic molecules (*CCL20, CCL22, CX3CL1, CXCL5, CX3CR1, CXCR3,* and *CD200*), cytokines and receptors (*IL27, TNF-α, IFN-γ, IL1-R1, LTβR,* and *IL1-R2*), transcription factors (*IRF4, STAT4,* and *FoxA1*), immunosuppression

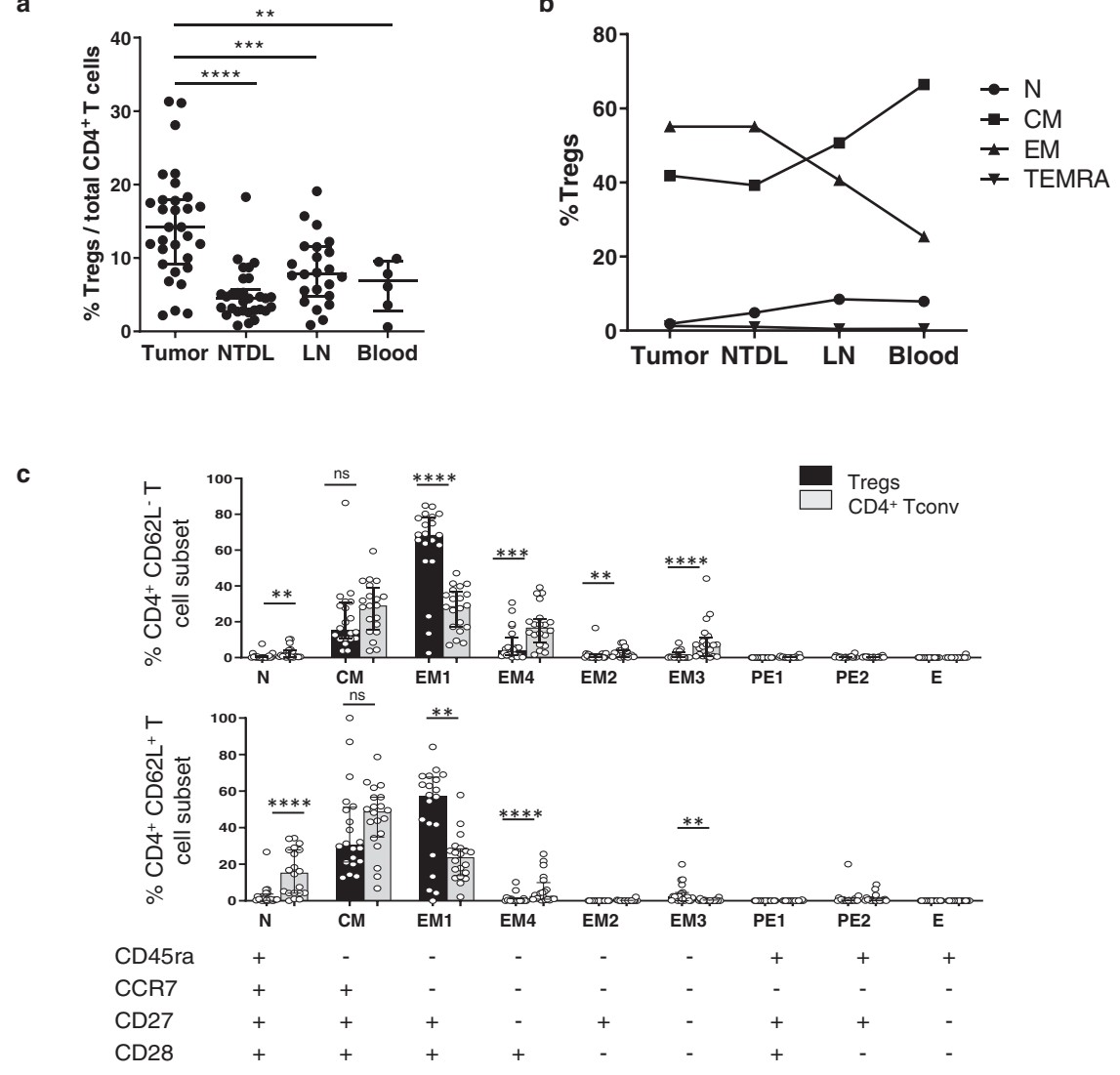

**Fig. 3 Stages of differentiation of Tregs according to their anatomical location. a** Frequency of CD3+CD25bright FoxP3bright cells among total CD4+
T cells in tumor (*n* = 31), non-tumoral distant lung (NTDL, *n* = 30), lymph nodes (LN, *n* = 23), and peripheral blood (*n* = 6) of NSCLC patients. Data are
represented as median with interquartile range. Wilcoxon–Mann–Whitney *U* test; **P = 0.0045, ***P = 0.0006, ****P < 0.0001. **b** Representation of the
main four stages of differentiation of Tregs in the different anatomical sites (*n* = 21 patients). **c** Stages of differentiation of both CD3+CD25bright FoxP3bright
Tregs (in black) and CD4+ Tconv (in gray) according to the differential expression of CD45ra, CCR7, CD27, and CD28 in TLS (CD62L+, top panel) and
non-TLS (CD62L-, bottom panel) areas (*n* = 21 patients). Data are represented as median with interquartile range. Wilcoxon–Mann–Whitney *U* test;
**P < 0.01, ***P < 0.001, ****P < 0.0001. CM, central-memory T cells, E exhausted T cells, EM effector-memory T cells, LN lymph node, N naïve cells, ns
non-significant, NTDL non-tumoral distant lung, TEMRA terminally differentiated effector-memory CD45ra+ T cell, TLS tertiary lymphoid structures.

molecules (*IL-10, CD39*, and *GARP*), and cytotoxic molecules
(*granzyme B, granulysine* (GNLY), and *FasL*). The differential
gene expression profile was less pronounced between Tregs
infiltrating tumors *versus* other distant sites such as LN and
NTDL with only 21 and 11 genes, respectively (Supplementary
Fig. 4). Moreover, most of them belong to the tumor *versus* blood
signature.

We confirmed the differential expression pattern of Tregs at
the protein level (Fig. 4b–d, Supplementary Fig. 3b–d). Again,
tumor and blood Tregs showed the most important difference
with 8% and 80% of cells that did not express any of CD38,
CD40L, CD69, and GITR molecules, respectively (Fig. 4b). This
difference was similarly observed for the other activation
(Fig. 4c) and ICP (Fig. 4d) proteins with 26 versus 65% and
10 versus 56% of Tregs expressing none of them in tumors
versus blood samples, respectively. The percentage of Tregs

positive for one molecule of interest was relatively homo-
geneous between sites, whatever the molecule considered. As a
consequence, the frequency of Tregs co-expressing more than
one marker dramatically increased from blood to tumors, with
common sequential events in all anatomical sites, i.e., first
CD69 (except for blood with a preferential CD38 expression),
then GITR, CD38, and lastly CD40L for the activation markers
(Fig. 4b); ICOS, then OX-40, and finally 4-1BB for the other
activation molecules (Fig. 4c); and TIGIT, followed by CTLA-4,
then TIM-3, and finally PD-1 or LAG-3 for ICP molecules
(Fig. 4d).

Altogether, Tregs have a distinct expression pattern of
activation and inhibitory receptors in tumors compared with
LN and blood. Most TIL-Tregs express both activation and ICP
molecules, whereas circulating blood Tregs exhibit a resting
phenotype in NSCLC patients.

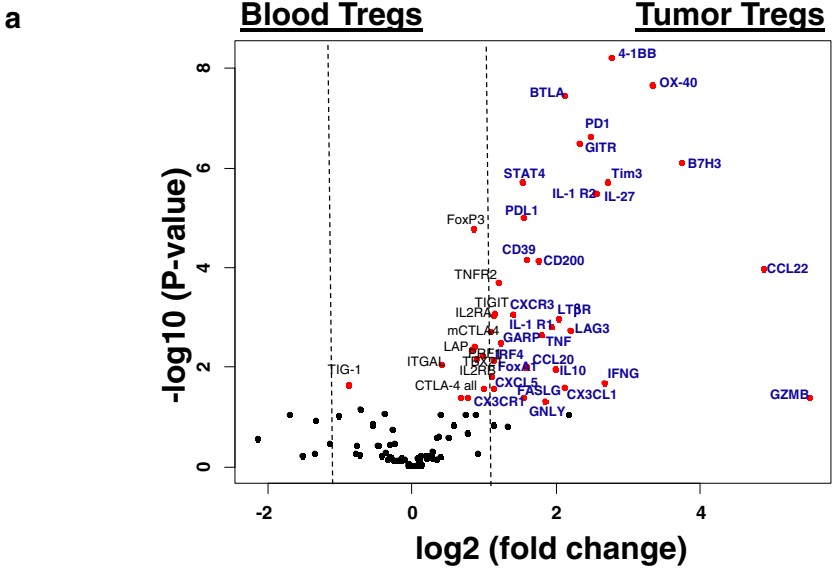

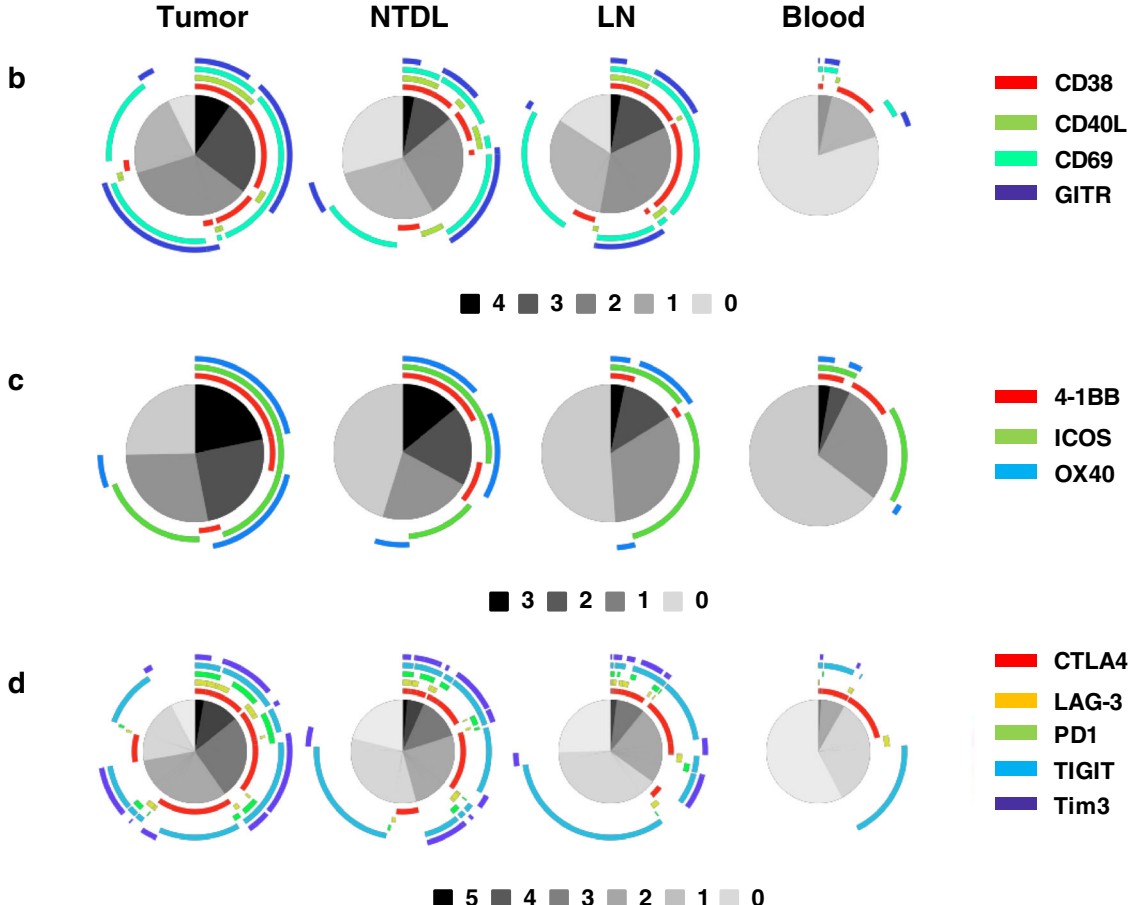

**Selective molecular and protein patterns of intra-tumoral Tregs *versus* CD4$^+$ Tconv in TLS and non-TLS areas**. The specific differentiation stage of Tregs observed in TLS versus non-TLS led us to investigate their activation and ICP status at the molecular ($n = 169$) and protein ($n = 14$) level, and to compare their phenotype to the CD4$^+$ Tconv cells. Along with the *FoxP3*, *IL2Rα*, and *IL2Rβ*, TLS and non-TLS Tregs significantly over-expressed many genes in comparison to the TLS and non-TLS

CD4$^+$ Tconv (Fig. 5a, b). For example, we observed for both Treg subsets an overexpression of some transcription factors (*FoxA1* for non-TLS Tregs, *Helios* (or *IKZF2*), and *IRF4* for both Treg subsets), chemokines and receptors (*CXCR3* for TLS Tregs and *CCL22*, *CCR4*, *CCR8* for both), cytokines and receptors (*IL10*, *IL27*, *IFN-α*, *TNFR2*, *IL1R1*, and *IL1R2* for both), activation molecules (*4-1BB* for non-TLS Tregs, and *GITR*, *ICOS*, *OX-40*, and *RANK-L* for both), several ICP molecules (*Tim-3* for TLS Tregs, *B7-H3* (*CD276*)

**Fig. 4 Selective molecular and protein pattern of Tregs infiltrating tumors *versus* non-tumoral sites.** The volcano plot (**a**) shows the genes over-expressed in Tregs isolated from blood *versus* tumors ($n = 20$ NSCLC patients). The *X*-axis shows the log2 fold change values for each gene, and the *Y*-axis shows the −log10 (*P*-values). The *P*-value < 0.05 (Student's *t*-test) is considered significant and genes showing the significant differential expression are highlighted in red dots. The red dots showing log2 fold change values higher than 1.2, are considered as significantly differentially expressed genes and are highlighted by their gene name in blue color. The percentage of Tregs expressing the combination of none to all different activation markers (CD38, CD40L, CD69, and GITR; panel **b**), other activation markers (4-1BB, ICOS, and OX40; panel **c**), and ICP molecules (CTLA-4, LAG3, PD1, TIGIT, and Tim3; panel **d**) is shown by each pie chart for different tissues ($n = 18$ patients). The black and gray colored portions of the pie chart represent the different combinations of the markers expressed by the proportion of Tregs ranging from 5 (expression of all markers; black color) to 0 (no expression of any marker; lightest gray color) marker. Each colored arc represents a single marker and thus, different colored arcs at the top of the pie chart show the distribution of the cells expressing different markers in the combinations. LN lymph node, NTDL non-tumoral distant lung.

for non-TLS Tregs, and membranous and soluble *CTLA-4* (*mCTLA* and *sCTLA*), *LAG-3, TIGIT* for both) and other molecules such as *CD39* and *GARP*. In accordance with the gene expression pattern, the percentage of Tregs positive for GITR, ICOS, 4-1BB, OX-40, CTLA-4, Tim-3, and TIGIT at the protein level was remarkably higher than in CD4$^+$ Tconv, whereas PD-1 expression was higher in CD4$^+$ Tconv than in Tregs (Fig. 5d). As for Th1 T cells, most Tregs express CXCR3 protein but only at an intermediate level (Fig. 5e). No statistical difference was observed regarding LAG-3 and CD40L positivity between the two T cell subsets.

Because TLS is considered as a privileged site for T cell activation, we next compared the gene expression profile of Tregs according to the TLS presence. Tregs in versus out of TLS exhibited a similar immune profile as very few genes were overexpressed on a given subset among the 120 genes analyzed (Fig. 5c). Among them, we can notice an overexpression of *CD62L* (or *SELL*) in TLS, and many ICP (*B7-H3* (or *CD276*), *GITR, PD-1, PD-L2*) and cytokines (*TNF-α, CCL22* and *IFN-γ*) in non-TLS, suggesting that non-TLS Tregs may exhibit a more immunosuppressive capacity compared to TLS-Tregs.

In conclusion, Tregs exhibit a distinct molecular pattern including activation and ICP molecules in comparison to CD4$^+$ Tconv in TLS and non-TLS areas of the tumor.

**TIL-Tregs inhibit the proliferation of autologous CD4$^+$ Tconv.** To determine the immunosuppressive capacity of TIL-Tregs, these cells were sorted and co-cultured with autologous CD4$^+$ Tconv isolated from blood and tumor of NSCLC patients. CD4$^+$ Tconv cells alone and stimulated with anti-CD3/anti-CD28 antibodies showed a strong proliferative capacity, demonstrating their ability to proliferate ex vivo (Fig. 6a). When TIL-Tregs were added to the culture, the proliferation of CD4$^+$ Tconv was strongly decreased in a ratio-dependent manner. Moreover, the secretion of IFN-γ and IL-2 by blood and TIL-CD4$^+$ Tconv was also drastically inhibited by the addition of TIL-Tregs (Fig. 6b).

Based on the profile of ICP expression by TIL-Tregs (Fig. 5d), neutralizing antibodies were added to the co-culture with CD4$^+$ Tconv. Anti-GITR and anti-CTLA-4 antibodies had no direct impact on the proliferation of CD4$^+$ Tconv cultivated alone, indicating that these antibodies have no direct effect on this population. However, the proliferation of TIL-CD4$^+$ Tconv was totally recovered when co-cultivated with Tregs and antibodies against GITR and CTLA- 4 (Fig. 7a, same effect was observed with blood and lymph node CD4$^+$ Tconv). Finally, no effect was observed on CD4$^+$ Tconv during the co-culture with antibodies against ICOS, PD-1, TIGIT and Tim-3 (Fig. 7b).

All together, these results demonstrate that TIL-Tregs exert an immunosuppressive activity on CD4$^+$ Tconv which can be recovered by blocking GITR or CTLA-4 axis.

**High density of TIL-Tregs is associated with a short-term survival, and negatively impacts the prognostic value of TLS.** Based on previous observation on the positive prognostic value of

TLS and effector CD8$^+$ T cells on the survival of NSCLC patients[4,7], we aimed to determine the prognostic importance of TIL-Tregs (i.e., total Tregs on the whole tumor section) according to these two variables. First, we observed that a high density of TIL-Tregs correlated with a poor outcome (median OS = 51 and 95 months for Tregs$^{High}$ and Tregs$^{Low}$ patients, respectively, $P = 0.004$, Fig. 8a) whereas high densities of TLS-mature DC and CD8$^+$ T cells were associated with long-term benefit (median OS = 82 and 35 months for TLS-DC$^{High}$ and TLS-DC$^{Low}$ patients, respectively, $P < 0.0001$, Fig. 8b; median OS = 69 and 35 months for CD8$^{High}$ and CD8$^{Low}$ patients, respectively, $P < 0.0001$, Fig. 8c). The combination of Tregs and TLS-DC - or Tregs and CD8$^+$ T cells—was a better predictor of survival than any individual variable (Fig. 8d, e). Not only Tregs$^{Low}$ TLS-DC$^{High}$ group identified patients having the longest survival (median OS not reached) but also Tregs$^{High}$ TLS-DC$^{Low}$ group identified patients having the worst outcome (median OS = 25 months) compared with each individual parameter. The same results were observed with Tregs$^{Low}$ CD8$^{High}$ and Tregs$^{High}$ CD8$^{Low}$ groups having the best (median OS not reached) and the worst outcomes (median OS = 35 months), respectively. Tregs$^{High}$ TLS-DC (or CD8)$^{High}$ and Tregs$^{Low}$ TLS-DC(or CD8)$^{Low}$ patients were at intermediate risk of death.

The best stratification was observed when the three markers were combined (Fig. 8f). Indeed, it better concentrates patients having the longest outcome (126 patients in Tregs$^{Low}$ TLS-DC$^{High}$ CD8$^{High}$ group with a median OS not reached) and shortest outcome (36 patients in Tregs$^{High}$ TLS-DC$^{Low}$ CD8$^{Low}$ group with a median OS of 27 months) out of 338 NSCLC patients.

A similar observation was made when Tregs were selectively counted in TLS. High density of TLS-Tregs was associated with poor outcome (Fig. 9a), and when combined with TLS-DC (or CD8$^+$ T cells), the best clinical outcome was for TLS-Tregs$^{Low}$ TLS-DC$^{High}$ (or TLS-Tregs$^{Low}$ CD8$^{High}$) patients (Fig. 9b, c). As for total Tregs, TLS-Tregs$^{Low}$ TLS-DC$^{High}$ CD8$^{High}$ group had the best outcome (median OS not reached) and TLS-Tregs$^{High}$ TLS-DC$^{Low}$ CD8$^{Low}$ group had the worst survival (median OS = 22 months) (Fig. 9d). Similarly, high CD8$^+$ T cells-to-Tregs ratio (Fig. 8g) or high CD8$^+$ T cells-to-TLS Tregs ratio (Fig. 9e) correlated with long-term survival of patients indicating that the balance of effector T cells to regulatory T cells is critical for the clinical outcome of patients.

Factors selected to determine associations with patient prognosis were immunology data, NSCLC disease stages, sex, age, and operation type (lobectomy or pneumonectomy). All satisfied PHA as well as log-linearity assumption (for continuous data). Both immunological data and NSCLC disease stages were significantly associated with patient prognosis in unadjusted and adjusted CPH models (Table 1). Furthermore, as NSCLC disease stages and immunological data were significantly associated one to another at 5% threshold (Chi-square test *p*-value = 0.042), NSCLC disease stage was deemed as a confounding factor in the

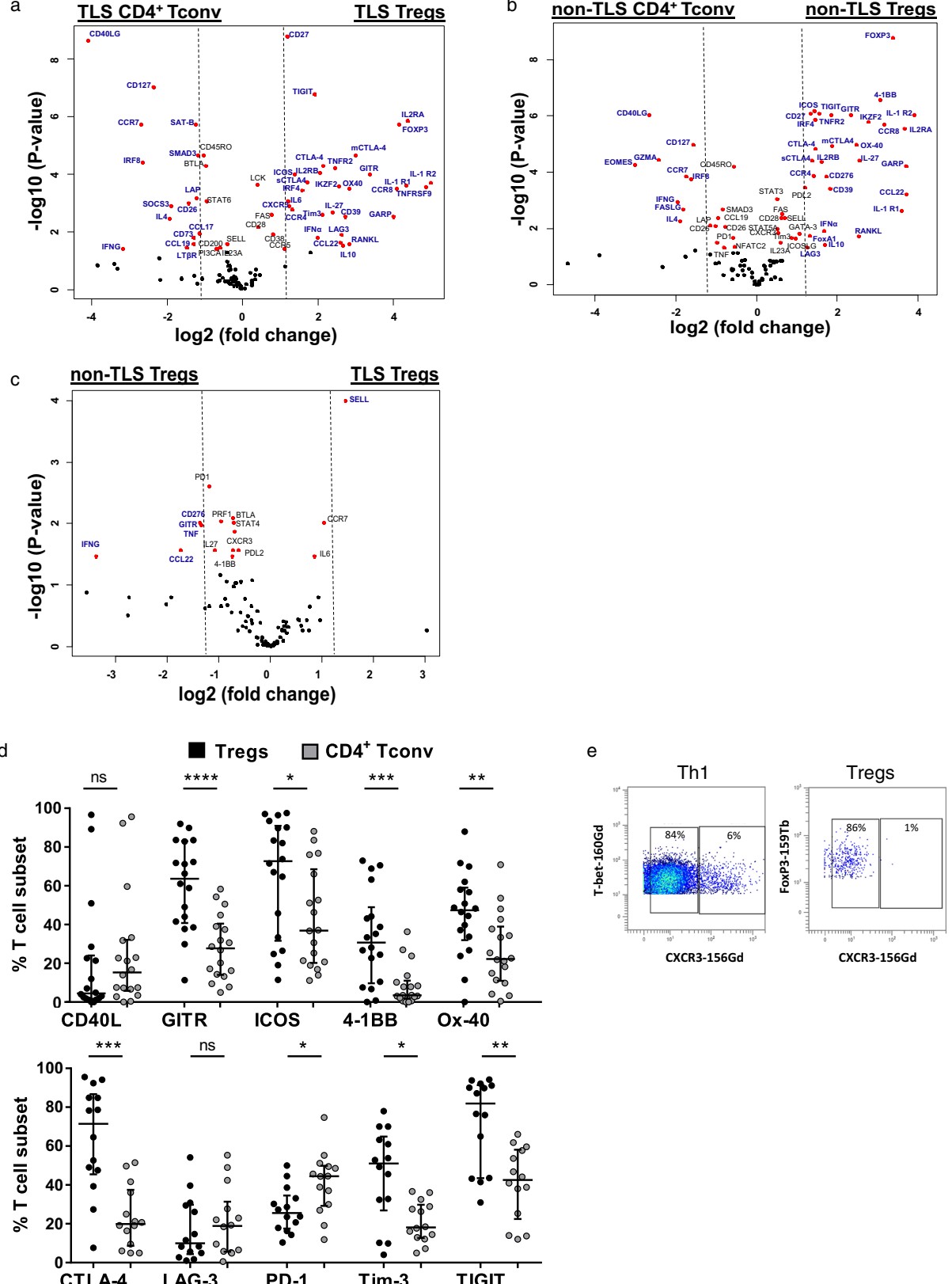

adjusted CPH model. Significant bad prognosis factors were stages IIIA (HR (95% CI) = 3.15 (1.45–6.84) compared to stage IA), Treg$^{Hi}$ TLS-DC$^{Lo}$/CD8$^{Lo}$ (HR (95% CI) = 4.00 (1.08–14.84) compared to 3 Hi) and interaction between stage IIA and 3 Lo (HR (95% CI) = 13.53 (1.75–104.50) compared to stage IA 3 Hi). Forest plot of adjusted hazard ratios is available in Supplementary

Fig. 5. Sensitivity analysis with Cox Lasso approach led to identical variable selection.

In conclusion on one hand, Tregs including TLS-Tregs and, on the other hand, TLS-DC and CD8$^+$ T cells have a dual impact on the outcome of NSCLC patients. And the combination of the three parameters provides a strong prognostic indicator of

**Fig. 5 Selective molecular and protein pattern of Tregs *versus* CD4⁺ Tconv infiltrating TLS and non-TLS areas of NSCLC tumors. a–c** The gene expression in terms of log2 fold change values for the sorted Tregs and CD4⁺ Tconv cells from TLS and non-TLS areas of lung tumors ($n = 20$ patients). The volcano plot shows genes over-expressed in **a** sorted TLS-Tregs (right) in comparison to sorted TLS-CD4⁺ Tconv (left), **b** non-TLS Tregs (right) *versus* sorted non-TLS CD4⁺ Tconv (left), and **c** TLS-Tregs (right) *versus* non-TLS Tregs (left) in lung tumors. The X-axis shows the log2 fold change values for each gene, and the Y-axis shows the −log10 (P-values). The P-value < 0.05 (Student's t test) is considered significant, and genes showing a significant differential expression are highlighted in red dots. The red dots showing log2 fold change values higher than 1.2, are considered as significantly differentially expressed genes and are highlighted by their gene name in blue color. **d** Percentage of Tregs (black circles) and CD4⁺ Tconv (gray circles) expressing activation markers ($n = 18$ tumors) and ICP ($n = 14$ tumors) at the protein level. **e** Illustration of the expression of CXCR3 on tumor-infiltrating CD3⁺ CD4⁺ CD8⁻ CD127⁺ T-bet⁺ Th1 T cells and CD3⁺ CD4⁺ CD8⁻ CD127⁻ FoxP3^bright CD25^bright Tregs. Data are represented as median with interquartile range. Wilcoxon–Mann–Whitney U test; *P < 0.05, **P < 0.01 ***P < 0.001, ****P < 0.0001.

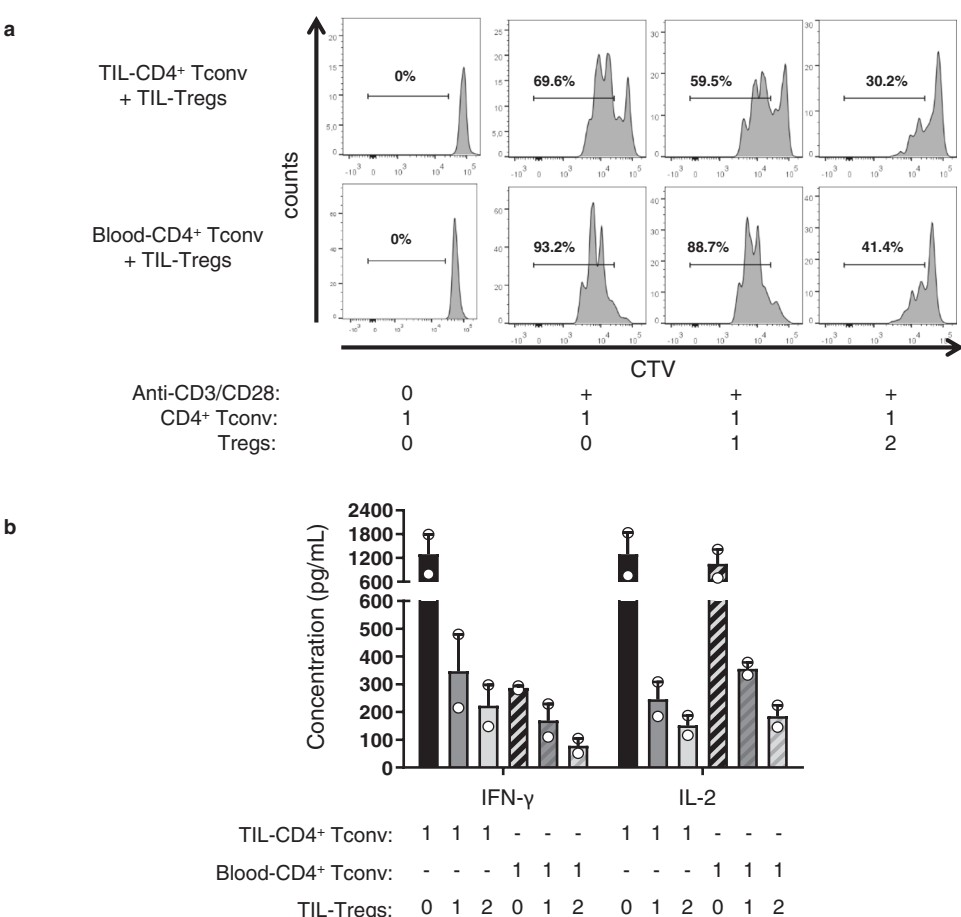

**Fig. 6 TIL-Tregs have a strong capacity of suppressing CD4⁺ Tconv proliferation ex vivo. a** Sorted CD2⁺CD8⁻CD4⁺CD127⁺ Tconv stained with cell trace violet (CTV) were cultured with or without sorted CD2⁺CD8⁻CD4⁺CD25^bright CD127⁻ Tregs for 72 hours at different ratios Tregs:CD4⁺ Tconv (0:1, 1:1 and 2:1) in presence of anti-CD3/anti-CD28 coated beads (1 bead/10 cells). Proliferation of CD4⁺ Tconv was measured by analyzing CTV dilution during cell culture. Representative data of one out of five experiments. **b** IFN-γ and IL2 secretion secreted by CD4⁺ Tconv previously sorted from tumors or blood of NSCLC patients and cultured with anti-CD3/anti-CD28 coated beads with or without autologous TIL-Tregs. The histograms represent the mean values +/− SEM of cytokine production of two independent experiments.

survival. It is thus a potential tool for identifying lung cancer patients at the highest risk of death.

## Discussion

The prognostic relevance of Tregs in different solid cancers has always been a matter of debate according to the stage and histological type of the tumor[15]. Moreover, based on their localization in tumors, Tregs may either suppress or potentiate the anti-tumor responses leading to a good[26], bad[16] or no association[27] with the survival of cancer patients. Thus, these controversies led us to speculate that Tregs may have different roles and phenotypes in different areas of the. With the advanced techniques available, we re-addressed the prognostic value of Tregs as a total

population and depending on their localization in the different sub-areas of lung tumors.

We observed that CD3⁺ FoxP3⁺ Tregs are prevalently distributed in the stroma and tumor-induced TLS but are rarely found in direct contact with tumor cells. We confirmed their phenotype (CD3⁺ CD4⁺ FoxP3^bright CD25^bright CD127⁻ cells) by flow cytometry, as mentioned in the literature[25,28]. High infiltration of the Tregs in lung tumors (compared with non-tumoral sites) is reflected by the high expression of the chemokine receptors CCR8 and CCR4 by TIL-Tregs. Importantly, CCR4 and its ligands CCL17 and CCL22 produced by TLS mature DCs, macrophages or tumor cells were shown to be involved in the Treg recruitment to the inflamed sites[16,22,29].

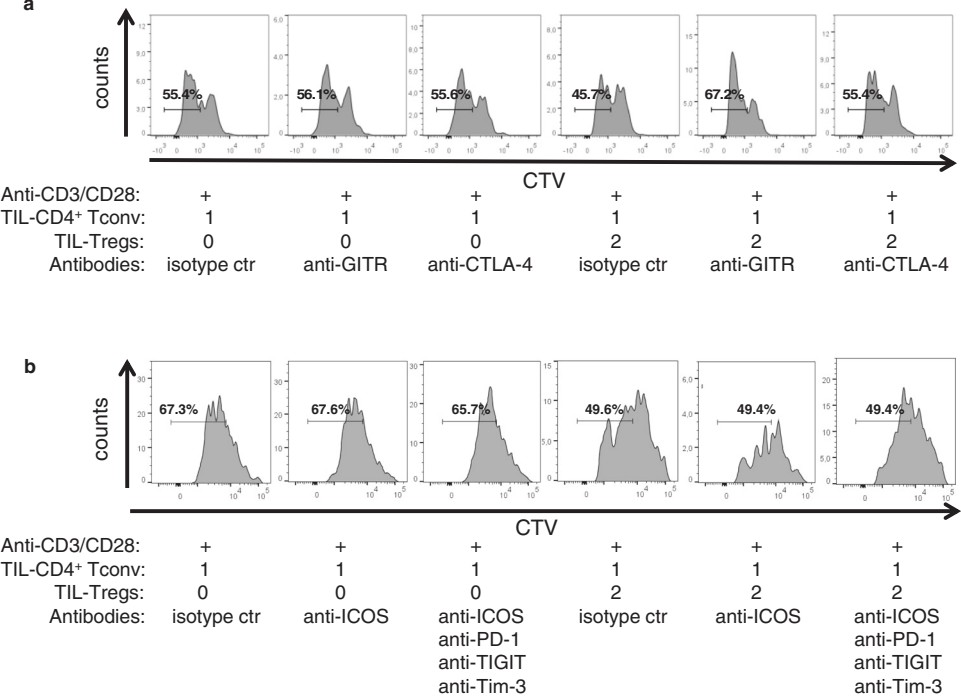

**Fig. 7 Impact of ICP blockade targeting Tregs on the proliferation of CD4+ Tconv.** The proliferation of sorted CD2+CD8−CD4+CD127+ TIL-CD4+ Tconv was measured by analyzing CTV dilution during cell culture with anti-CD3/anti-CD28 coated beads (1 bead/10 cells) with or without autologous sorted CD2+CD8−CD4+CD25bright CD127− TIL-Tregs with or without neutralizing antibodies against CTLA-4 and GITR (**a**), and ICOS, PD-1, TIGIT, and Tim-3 (**b**).

Spatial localization of Tregs in different areas of tumors led us to study their special phenotypic status. Therefore, we evaluated the phenotype of Tregs based on their presence in TLS and non-TLS areas. We observed that TIL-Tregs predominantly show CM and EM phenotypes. Furthermore, the differentiation status of Tregs was distinct between tumor *versus* blood or lymph node, but quite similar with NTDL. Advanced differentiation status led us to further study the activation status of Tregs. At both, protein, and molecular level Tregs expressed different activation and ICP markers, reflecting their functional status in tumor microenvironment. This also highlights that strategies targeting ICP may not act overall Treg population.

Tregs drive the immunosuppression through several mechanisms, including co-stimulatory and co-inhibitory molecules[14]. The role of the co-stimulatory molecules ICOS, OX-40 and GITR in the activation and immunosuppressive function of Tregs has been largely studied in the last few years[30–32]. Although, the role of GITR in Tregs has been controversial, it has been found that Tregs constitutively express GITR[33]. The TNF super family receptors TNFR2, 4-1BB, and OX-40 have additive role in their suppressive ability. Co-expression of GITR, OX-40, and TNFR2 along with TCR signaling has been found to favor the thymic differentiation of Tregs[34]. Tregs in lung tumors highly expressed GITR, 4-1BB, OX-40, TNFR2 along with high expression of CD25, suggesting their advanced differentiation and activation status. CD137 or 4-1BB displays tumor specificity across multiple cancer types[35]. Most interestingly, we observed that a proportion of Tregs express several ICP molecules with high levels of CTLA-4, TIGIT, Tim-3, B7-H3, PD1 and its ligands, in comparison to the blood resting Tregs. The expression of ICP itself is suggestive of their highly activated profile. In the literature, this expression has been found to be associated with Tregs function while interacting with other immune cells. For instance, CTLA-4 on Tregs and CD80/CD86 on DCs triggers the production of the IDO enzyme by DCs, which leads to induction of tolerance[36,37]. Moreover, TIGIT on Tregs has

been shown to be involved in the inhibition of pro-inflammatory responses by Th1 and Th17[38]. Furthermore, Helios+ memory Tregs expressing TIGIT and FCRL3 are highly suppressive Tregs[39,40]. In our study, we observed a very high gene expression of TIGIT and all forms of CTLA-4 by TIL-Tregs in TLS and non-TLS areas, suggesting their involvement in the immunosuppressive mechanisms in lung tumors. Interestingly, the remarkable success of the anti-CTLA-4 immunotherapy in cancer patients[41] may be the consequence of a dual effect of monoclonal antibody with the recovery of effector T cell function but also with the inhibition of Treg activity. With a confirmation of the high expression of the activation and ICP markers at the protein level and in comparison, to CD4+ Tconv cells, we highlight the exclusive phenotype of Tregs in lung tumors; even though the differentiation status of these two cell subsets was quite similar. These Tregs are strongly capable of suppressing the proliferation ex vivo of autologous CD4+ Tconv cells from tumors but also of those circulating in the blood, which are less influenced by the immunosuppressed environment. These results strongly support the immunosuppressive phenotype of TIL-Tregs.

In the absence of any infection, CCR7−/− mice spontaneously develop BALT (Bronchus-Associated Lymphoid Tissue) due to the lack of tolerance by Tregs[42]. In other mouse studies, it was observed that the depletion of Tregs increased the HEV formation, T cell infiltration, and tumor destruction[20,43]. A preclinical mouse model with lung adenocarcinoma demonstrated the immunosuppressive role of the Tregs in lung tumor-associated TLS[44]. This study showed that Treg depletion, in the lung tumor-bearing mice, improves the anti-tumor response and infiltration of tumor antigen-specific T cells in the tumor and induces the destruction of the tumors *via* a protective response generated in the TLS. Among cancer patient-associated studies, there is poor evidence showing the role of Tregs in shaping anti-tumor responses and especially their influence on TLS. For the first time, we showed that the

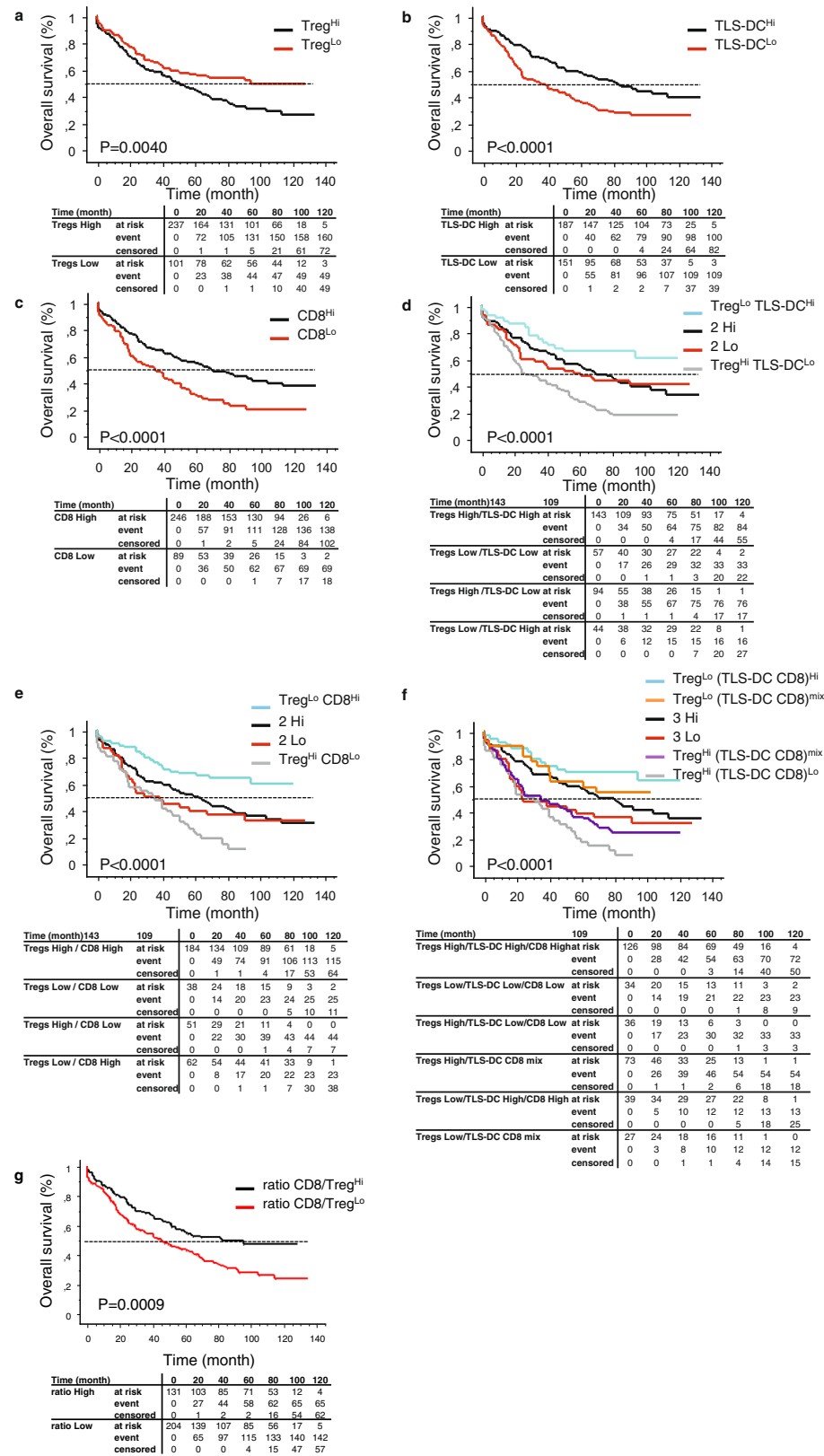

**Fig. 8 Overall survival of NSCLC patients according to the densities of tumor-infiltrating Tregs, TLS-DC, and CD8+ T cells.** Densities of total of CD3+ FoxP3+ Tregs (i.e., TIL-Tregs infiltrating the whole tumor section), TLS-DC-Lamp+ mature DC, and CD8+ T cells were determined using serial sections of tissues ($n = 338$ NSCLC). The Kaplan–Meier survival graphs were plotted for the determination of the OS of patients. The log-rank test was used to determine the statistical significance of the data. The patients were stratified into high and low groups according to cell densities and survival determined (**a**, Total Tregs; **b**, TLS-DC; **c**, CD8+ T cells; **d**, Tregs TLS-DC; **e**, Tregs CD8, **f**, Tregs (TLS-DC CD8); and **g**, total CD8-to-total Tregs ratio). The table below each Kaplan-Meier curve graph shows the number of patients at risk, number of events and censored according to the cell density group.

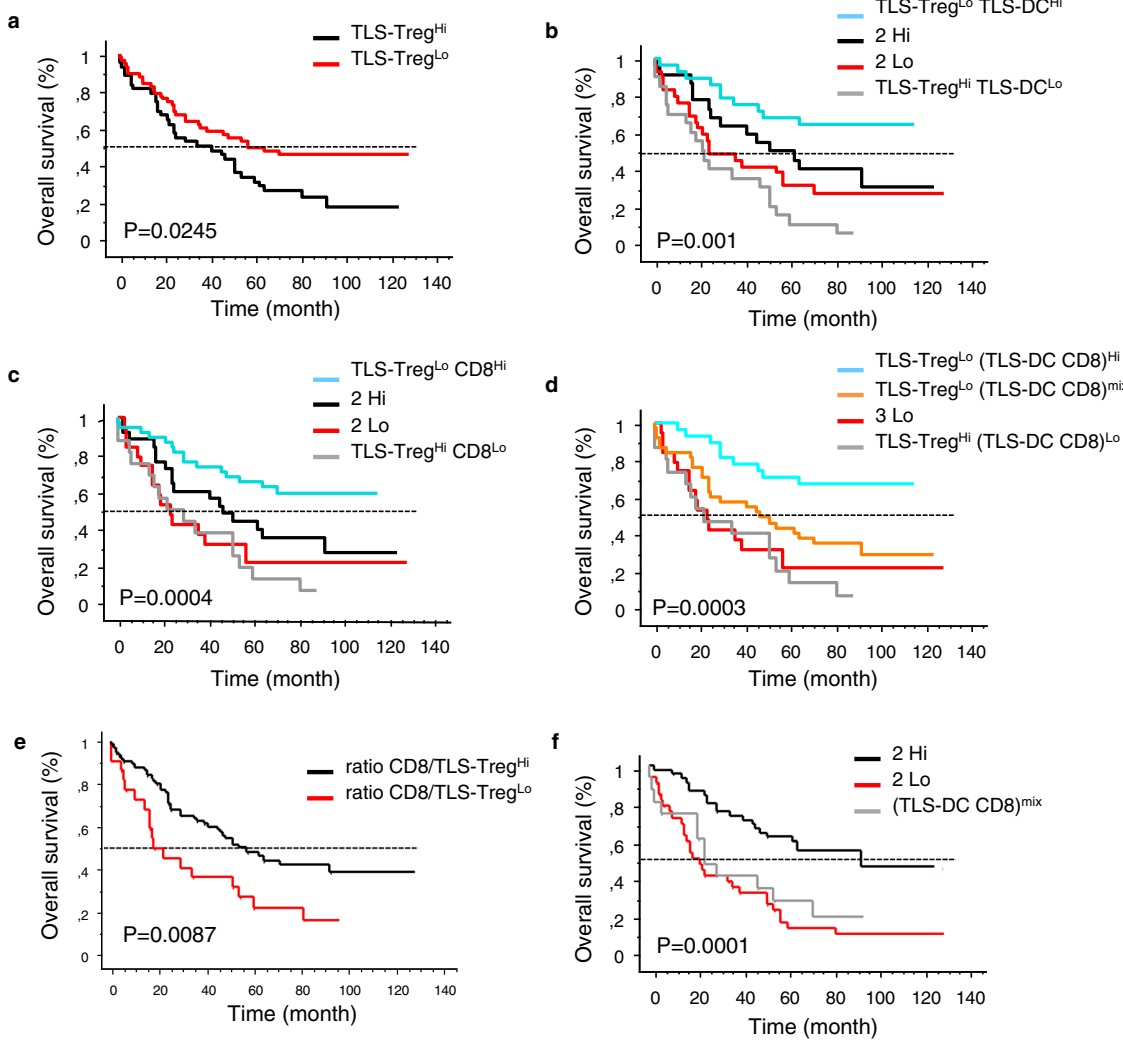

**Fig. 9 Kaplan–Meier curves for TLS-Tregs alone or in combination with TLS-DC and/or CD8+ T cells.** Densities of total Tregs, TLS-mature DC, and CD8+ T cells were determined by IHC using serial sections of FFPE lung tumor samples. The Kaplan–Meier survival graphs were plotted for the determination of the percentage OS of the patients. The Log-rank test was used to determine the statistical significance of the data. Overall survival of NSCLC patients according to densities of **a** TLS-Tregs, **b** TLS-Tregs TLS-DC, **c** TLS-Tregs CD8, **d** TLS-Tregs (TLS-DC CD8), **e** total CD8-to-TLS-Tregs ratio, **f** TLS-DC CD8.

immune profile of human Tregs in TLS and non-TLS is quite similar with, however, very interesting specificities. For instance, TLS Tregs overexpressed IL6, a cytokine known to drive the B cell and follicular helper T cell differentiation, and to support B cell production of IgG. Non-TLS Tregs overexpressed some ICP including GITR, PD1, PDL2, which may be critical for a crosstalk with tumor cells and effector T cells, respectively. This ICP profile (GITR, PD1, PD-L2 overexpress by non-TLS Tregs versus CTLA-4 express by both Treg subsets) is of particular interest regarding response to therapy targeting alone or in combination these ICP. IFN-γ is known to induce the expression of Th1 chemokines like CXCL-9, −10 and −11, as well as their shared receptor CXCR3. Thus, secretion of IFN-γ by non-TLS Tregs may favor the recruitment of CXCR3+ cells along with Tregs and Th1 cells, and ultimately abolish Th1 immune response. Of note, the overexpression of perforin by non-TLS Tregs may also limit the onset of anti-tumor immunity by directly killing effector T cells. Frafjord et al. reported the predominance of Tregs (and Th2) *versus* Th1 (cells per mm² [45]) while we have previously reported that the TLS presence correlates with a specific intra-tumoral immune

contexture characterized by the overexpression and the coordination of genes related to Th1 orientation, T cell activation and cytotoxic effector functions while no correlation was observed between TLS and Th2 signature or immunosuppression-related genes[7]. These apparent discrepancies could be due to the limited number of NSCLC patients included in the Frafjord's study.

In breast cancer patients, the presence of Tregs in the lymphoid aggregates is correlated with the poor survival, whereas they were not associated with prognosis in the other areas of the tumor[16]. As observed in prostate cancer and lung metastasis[46,47]. We showed that the presence of a high number of Tregs in TLS and the whole tumor negatively impacts the survival of patients. In lung cancer patients, we have already demonstrated the presence and favorable role of TLS in terms of longer survival of patients[4,6,7]. Using this observation, when we combined Tregs density with that of DC-Lamp+ mature DC or CD8+ T cells, we achieved a better stratification and could identify a group with the highest survival. Moreover, Tregs appear to be an important prognosticator in a group of patients with the highest risk of death. In accordance with the literature, we observed that the

**Table 1 Hazard ratio (HR) and 95% Confidence Interval (95%CI) for overall survival of NSCLC patients ($n = 338$).**

|  | Number of events | HR (95% CI)[a] | HR (95% CI)[b] |
|---|---|---|---|
| Sex (ref. F) | 169 | 1.43 (1.02–2.02)* | 1.38 (0.95–2.01) |
| Age | 209 | 1.02 (1.00–1.03)* | 1.01 (1.00–1.03) |
| Histological type (ref. adenocarcinoma) |  |  | / |
| Squamous cell carcinoma | 68 | 1.01 (0.75–1.36) |  |
| Large cell carcinoma | 10 | 1.26 (0.66–2.40) |  |
| Carcinomas with pleomorphic, sarcomatoid, or sarcomatous elements | 4 | 1.72 (0.63–4.66) |  |
| Operation type (ref. lobectomy) | 41 | 2.18 (1.55–3.07)*** | 1.61 (1.08–2.41)* |
| NSCLC disease stage (ref. IA) |  |  |  |
| IB | 52 | 1.35 (0.82–2.20) | 1.75 (0.80–3.85) |
| IIA | 20 | 1.07 (0.59–1.96) | 0.84 (0.32–2.19) |
| IIB | 35 | 2.17 (1.28–3.68)** | 1.74 (0.67–4.54) |
| IIIA | 73 | 3.28 (2.05–5.25)*** | 3.15 (1.45–6.84)* |
| IIIB + IV | 6 | 2.77 (1.13–6.83)* | 1.76 (0.45–6.82) |
| Smoking status (in pack years) | 195 | 1.00 (1.00–1.00) | / |
| Stratification groups into high and low groups according to cell densities (ref. 3 Hi) |  |  |  |
| 3 Lo | 23 | 1.48 (0.92–2.36) | 1.00 (0.22–4.65) |
| Treg[Hi] TLS-DC/CD8[mix] | 55 | 1.76 (1.24–2.51)** | 1.62 (0.62–4.21) |
| Treg[Hi] TLS-DC[Lo]/CD8[Lo] | 32 | 2.52 (1.66–3.85)*** | 2.12 (1.25–3.60)* |
| Treg[Lo] TLS-DC/CD8[mix] | 12 | 0.76 (0.41–1.40) | 0.00 (0.00–Inf) |
| Treg[Lo] TLS-DC[Hi]/CD8[Hi] | 13 | 0.50 (0.28–0.90)* | 0.47 (0.06–3.68) |
| Interactions between NSCLC disease stage and immunology data |  |  |  |
| Stage IIA x 3 Lo |  | 17.93 (2.33–137.74)** | 13.53 (1.75–104.50)* |

CPH models were performed to identify prognosis factors for OS in univariate and multivariate settings. The patients were stratified into high and low groups according to cell densities and survival determined.
From CPH model, *$P < 0.05$, **$P < 0.01$, ***$P < 0.001$.
[a]Unadjusted.
[b]Adjusted on sex, age, operation type (lobectomy or pneumonectomy), NSCLC disease stage and immunological data (and interaction term).

ratio of the $CD3^+$ T cells (or mature DC or $CD8^+$ T cells) with Tregs (whole tumor or in TLS) was a stronger prognosticator than each variable alone.

In summary, the high density of Tregs in the whole tumor and including that in TLS, is associated with the reduced survival of NSCLC patients. ICP blockade, an emerging tool for immunotherapy is also showing promising results in NSCLC[48,49]. Thus, exploiting markers like CTLA-4, TIGIT, OX-40, 4-1BB and GITR, which we observed to be highly expressed by TIL-Tregs, can be a potential therapeutic target for the treatment of the lung cancer patients.

## Methods

**Patients.** Primary lung tumor samples were obtained from NSCLC patients operated at Institut Mutualiste Montsouris, Hotel Dieu, and Cochin hospitals (Paris, France). A retrospective cohort of 338 NSCLC patients operated between the years 2001 to 2005 was enrolled in this study. Pathologic staging of lung cancer was determined according to the new TNM staging classification[50]. Histological subtypes were determined according to the classification of the WHO[51]. Among the retrospective cohort of patients, the patients treated with neoadjuvant chemotherapy and radiotherapy were excluded. The time between the surgery and the last follow-up or death is considered as the observation time for this cohort. The data on long-term outcomes were obtained after interaction from municipality registers or the family of the patient.

Fresh tumor biopsies, non-tumoral distant lung (NTDL), lymph nodes (LN) specimens, and blood were also obtained from 60 NSCLC patients undergoing surgery (prospective cohort). Samples were obtained from patients with a written consent and by a protocol that was approved by the local ethic committee (no. 2008-133, no. 2012-0612 and no. 2017-A03081-52) of the European Georges Pompidou hospital and Institut Mutualiste Montsouris (Paris, France) an application with the article L.1121-1 of French law. The main clinical and pathological features of the two cohorts are presented in Supplementary Table 1.

**Immunostaining and cell quantification.** Formalin fixed, paraffin-embedded tissue serial sections with 5 μm thicknesses were used for immunohistochemistry (IHC) double staining and multiplex-immunofluorescence (IF) staining. Briefly, tissue sections were deparaffinized, rehydrated and treated with the antigen retrieval buffer. The antigen retrieval was performed using TRS buffer (Dako, France) at 97 °C for 30 min in water bath, in case of the CD3/FoxP3 and CD62L/FoxP3 double staining. The sections were incubated in the 3% $H_2O_2$ and ready-to-use protein bloc (Dako, France) solution for 30 min before the addition of the appropriate primary and secondary antibodies. The antibodies and reagents are listed in Supplementary

Table 2. The enzymatic activity was performed using the AEC (3-amino-9-ethyl-carbazole) kit and APS (Vector blue alkaline phosphatase) substrate kits. Images were acquired using Nanozoomer (Hamamatsu) with NDPview software. The same pre-treatments were used for the 5 plex-immunofluorescence staining. Tissue sections were successively stained with the appropriate primary antibody (CD3, CD20, CD62L, FoxP3, DAPI) and HRP-coupled secondary antibodies. The sections were then washed three times for 5 min and then incubated with the fluorophore tyramide reagent that covalently binds the tissue. Finally, a stripping step was performed by heating slides for 10 min at 90 °C before proceeding to another staining cycle. The specificity of each antibody was tested by using isotype control antibodies. The effectiveness of the stripping step was checked for each multiplex staining. Nucleus staining was performed by using ProLong® Gold with DAPI (Molecular Probes). IF slides were scanned with an Axio Observer microscope (Zeiss) and analyzed by Zen (Zeiss) and Halo (Excilone) software.

*Cell quantification.* The area of interest was manually drawn on the scanned images of tumor tissue sections. The surface area of this region of interest was also determined using the Calopix software (Tribvn, France). $CD3^+$ total T cells and $CD3^+FoxP3^+$ T cells were quantified in the whole tumor section using the same software and expressed as a number of cells/mm² (number of total cells divided by the surface area in mm²). The region of TLS was determined manually by referring to the double staining DC-Lamp/CD3 to determine the TLS area. The DC-Lamp/CD3 staining was performed on the serial tumor sections from the same set of patients. The surface area of the TLS was also determined by the Calopix software. The density of $CD3^+FoxP3^+$ cells in TLS was determined with an automatic counting using Calopix. The quantification of the TLS-DC-Lamp$^+$ DC, $CD8^+$ T cells, was determined, as previously described[6,7].

**Flow cytometry.** A total of 34 NSCLC fresh tumor samples were enrolled in this study. Tumors and non-tumoral tissue specimens were mechanically dilacerated and digested in a non-enzymatic solution (cell recovery solution, BD Biosciences, France). The total mononuclear cells were obtained after a ficoll gradient. Mononuclear cells were stained with multiple panels of the fluorescently conjugated antibodies or their matched isotype controls (Supplementary Table 3). Further, cells were fixed and permeabilized using fixation/permeabilization kit (ebioscience, San Diego, CA) for intracellular staining of FoxP3. Then, the cells were washed, and acquired on the Fortessa cytometer (BD Biosciences, France). Data were analyzed using Flow Jo 9.7.6 (Tree Star Inc, Ashland, OR) and Spice 5.3.5 (developed by Mario Roederer, Vaccine Research Center, NIAID, NIH) software programs. The gating strategies are detailed in Supplementary Fig. 1.

**Cell sorting.** For the gene expression study, Tregs and Tconv subsets were sorted from 20 tumors and non-tumoral specimens by the in house designed protocol[21].

The combination of Easysep$^{TM}$ untouched human CD4$^+$ T cell kit (stem cell technologies, France) and flow cytometry cell sorting with the cocktail of antibodies (anti-CD2, CD4, CD8, CD25, CD127, and CD62L) was used to achieve the high purity of the cell subsets. Four populations of cells, namely TLS Tregs (defined as CD2$^+$CD4$^+$CD8$^-$CD25$^{hi}$CD127$^-$CD62L$^+$), non-TLS Tregs (defined as CD2$^+$CD4$^+$CD8$^-$CD25$^{hi}$CD127$^-$CD62L$^-$), TLS CD4$^+$ Tconv (Tconv defined as CD2$^+$CD4$^+$CD8$^-$CD25$^-$CD127$^+$CD62L$^+$) and non-TLS CD4$^+$ conventional T cells (Tconv defined as CD2$^+$CD4$^+$CD8$^-$CD25$^-$CD127$^+$CD62L$^-$) were then sorted from fresh tumor and non-tumoral tissue specimens ($n = 20$). The purity of cells achieved between 98–100% for all the cell subsets sorted from tumor and non-tumoral tissue specimens. The cells were sorted directly into vials containing the cell lysis buffer RLT + 10% β-mercaptoethanol in order to obtain the best quality and quantity of the total mRNA. The antibodies and reagents used are listed in Supplementary Table 3.

For the functional assay experiment, the CD4$^+$ Tconv cells and Tregs were sorted in PBS + 5% FCS (Fetal calf serum) using the same in house designed protocol. The cells were co-cultured in a 96-well round bottom plate in complete RPMI 1640 medium, containing 100 U/ml penicillin and 100 µg/ml streptomycin, 2 mM L-glutamine, 5 mM sodium pyruvate, 1 mM HEPES, 50 µM 2-mercaptoethanol (Gibco, France) and 10% AB Human serum (Eurobio, France). The medium was filtered and maintained in sterile conditions.

**RNA extraction, reverse transcription, and gene expression analysis**. Total mRNA from the sorted cells was extracted with the RNeasy micro kit (Qiagen, France) according to manufacturer's instructions, and RNA quantity and quality were determined using the 2100 Bioanalyzer (Agilent Technologies, France). The mRNA was reverse transcribed to cDNA using a superscript VILO kit (Life Technologies, France). The samples below 1 ng of mRNA were amplified by 9 cycles. When the quantity of mRNA was more than 1 ng of mRNA, it was amplified by 7 cycles of PCR using Taqman PreAmp 2x and MTE primers (NanoString technologies, Seatle, USA). Two specific probes (capture and reporter) for each gene of interest were applied. The customized reporter probe and capture probe code-set of selected 125 genes, including five housekeeping controls (β-actin, GAPDH, EEF1G, OAZ1, and RPL19) and cell lineage controls (CD3, CD4, CD8, CD19, CD138, and EpCAM) were used for the hybridization according to the manufacturer's instructions (Nanostring Technologies) (Supplementary Data 1). Water was used as a negative control to check the background noise. The hybridized samples were recovered using the NanoString Prep-station and the mRNA molecules counted with the digital nCounter. The number of counts represented the gene expressed. The high expression level of gene means the readout in terms of a high number of counts of that particular gene. The positive and negative controls and one patient's RNA sample as an internal control were used to check the technical consistency between each strip of experiments. Raw data are available in Supplementary data 2.

**Immunosuppression assay**. Tregs and CD4$^+$ Tconv cells were purified from the lung tumor and blood specimens ($3 \times 10^3$/well). CD4$^+$ Tconv cells were labeled with CTV (life technologies, France, working concentration: 10 µM), and then cultured with anti-CD3/anti-CD28 beads (1 bead/10 cells, Invitrogen) and without any exogenous cytokine in 96 well round bottom plates in complete RPMI 1640 with 10% AB$^+$ heat-inactivated human serum for 72 h at 37 °C. Autologous Tregs were co-cultured with CD4$^+$ Tconv cells at different ratios (0:1, 1:1 and 2:1). CD4$^+$ Tconv proliferation was measured on Day 3 by analyzing CTV dilution by flow cytometry. For some experiments, cells were cultured with neutralizing antibodies against anti-GITR (clone DT5D3, Miltenyi Biotec), anti-CTLA-4 (clone BN13, BioXCell), anti-ICOS (clone 314.8 kindly provided by Dr. D. Olive), anti-PD-1 (clone J116, BioXCell), anti-TIGIT (clone MBSA43, Affymetrix), and anti-Tim-3 (clone F38-2E2, Affymetrix), (working concentration at 10 µg/mL).

**Cytokine detection**. IFN-γ and IL-2 production were analyzed in the supernatant using the cytometric bead array kit for human Th1/Th2/Th17 (Becton Dickinson) according to manufacturer's instructions. Results were analyzed with FCAP array™ software V3.0 (Becton Dickinson).

**Statistics and reproducibility**. For the flow cytometry data, Wilcoxon–Mann-Whitney U test was used to compare the density of cells in the different tumors. For gene expression study and volcano plot demonstrations, the 'nSolver' (Nanostring Technologies) and R (CRAN) software were used. The raw data were normalized with an average count of the five housekeeping genes using the "nSolver" software. The Student T and ANOVA tests were used to compare the gene expression profile of the different groups. To avoid the inclusion of the false positive results, we computed the $P$-values with the false discovery rate (FDR) method. The data were represented as volcano plots.

The overall survival (OS) curves were estimated by the Kaplan–Meier method, and differences between groups of patients were calculated using log-rank test. Patients were stratified into two groups according to the high and low densities of immune cells using the minimum $P$-value approach, as previously published[6,7]. Briefly, this approach assesses the cut-off for the best separation of NSCLC patients referring to their clinical outcome. Optimal cut-off values are represented in

Supplementary Fig. 2. The optimal cut-off values are 21.93277 for CD3$^+$FoxP3$^+$ Treg cells/mm$^2$, 127.0348 for TLS CD3$^+$FoxP3$^+$ Treg cells/mm$^2$, 1021.976 for CD3$^+$FoxP3$^-$ Tconv cells/mm$^2$, 191.177 for CD8$^+$ T cells/mm$^2$, and 1.248 for TLS DC-Lamp$^+$ DC cells/mm$^2$ of tumor areas (Supplementary Fig. 2). Clinical and immunological parameters were also used for the statistical analyses. A multivariate Cox proportional hazard (CPH) model was also implemented in order to estimate relationships between the immunological data, patient prognosis and NSCLC disease stages. NSCLC disease stages IIIB and IV were gathered together in a single category. Prior variable selection was performed using a univariate modeling. Sensitivity analyses included alternatives for eventual violation of the proportional hazard assumption (PHA, tested via Grambsch & Therneau approach[52],) and a Cox Lasso approach[53] for the variable selection, using 10-folds cross-validation to determine hyperparameter value and to study impact on partial likelihood deviance and coefficients. Source data are available in Supplementary data 3. All the analyses were performed with Prism 5 (GraphPad), Statview (Abacus system) and R (http://www.r-project.org/) softwares.

**Reporting summary**. Further information on research design is available in the Nature Portfolio Reporting Summary linked to this article.

## Data availability
All data relevant to the study are included in the article or uploaded as supplementary data: supplementary data 1 (list of genes and accession number), supplementary data 2 (Figs. 4 and 5a–c), supplementary data 3 (Figs. 8 and 9), supplementary data 4 (Fig. 2a), supplementary data 5 (Fig. 3a), supplementary data 6 (Fig. 3b), supplementary data 7 (Fig. 3c), supplementary data 8 (Fig. 5d), supplementary data 9 (Fig. 6b).

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

## Acknowledgements

The authors thank Mr. Sylvain Leveugle for technical support in statistical analysis, and Dr. Hélène Froher-Ting, Dr. Estelle Devevre, and Dr. Christophe Klein (platform "Center of cellular imaging and cytometry", Cordeliers Research Center, Paris, France) for their excellent technical support and assistance in cell sorting, and cell quantification from digital slides, respectively. We also thank Dr. David Gentien and Mr. Bertrand Albaud (Genomic platform, Department of Translational Research of Curie Institute, Paris, France) for the technical support in the gene expression profiling. We thank Marie Laviron and Alexandre Boissonnas for both technical assistance and helpful scientific discussions. Finally, we thank Ms. P. Bonjour and Ms. Nathalie Jupiter for their help in tissue section. This work was supported by the "Institut National de la Santé et de la Recherche Médicale (INSERM), the "Institut National du Cancer" (INCa-DGOS_10888, Dieu-Nosjean), Sorbonne Université, and Université de Paris. P. Devi-Marulkar and Jérémy Goc were supported by a grant from the "Fondation ARC pour la Recherche sur le Cancer". Hélène Kaplon was supported by a grant from "La Ligue contre le Cancer".

## Author contributions

Concept and design: M.-C.D.N. Development of methodology: P.D.-M., S.F., P.K., J.G., C.G., H.K., S. Knockaert, J.M., S. Katsahian, M.L., and M.-C.D.N. Acquisition of data: P.D.-M., S.F., P.K., J.G., C.G., J.M., S. Knockaert, M.L., and M.-C.D.N. Analysis and interpretation of data: P.D.-M., S.F., J.G., C.G., D.O., J.M., S. Katsahian, M.L., and M.-C.D.N. Writing of the manuscript: P.D.-M., M.L., and M.-C.D.N. Review and/or revision of the manuscript: all co-authors. Administrative, technical, or material support: D.O., P.V., D.D., and M.A. Funding acquisition: M.-C.D.N. Study supervision: M.L. and M.-C.D.N.

## Competing interests

M.-C.D.-N. is listed as a co-inventor of four patents (including one with J.G., C.G., and P.D.-M. as co-inventors, too) regarding the method for the prognosis of survival time of a patient suffering from a solid cancer that is owned by Inserm and Inserm Transfert. D.O. is cofounder and shareholder of Imcheck Therapeutics, Emergence Therapeutics and Alderaan Biotechnology. The remaining authors declare no competing interests.

## Ethics approval and consent to participate

Written informed consent before their inclusion in the study was obtained from all patients. Protocols were approved by the local ethics and human investigation committees (nos. 2008-133, 2012-0612 and 2017-A03081-52), in the application of article L.1121-1 of the French Public Health Code.
