## [Peer Review File · Communications Biology]

Reviewers' comments:

Reviewer #1 (Remarks to the Author):

This manuscript investigates regulatory T cells (Tregs) in human NSCLC with a main focus on tertiary lymphoid structures (TLS). Some potentially interesting findings are reported.

Specific comments:

- 1) The text of the manuscript needs some serious improvements to improve its clarity. The current text is difficult to follow. The authors need to explain better what exactly they have done and found.
- 2) Several reports have previously shown an association between high numbers of Tregs in NSCLC tumors and worse prognosis (e.g.: PMID: 20234320, 21719142, 23269987). The authors need to clearly refer to this previous work in the Introduction and compare their new findings with these previous studies in the Discussion.
- 3) FOXP3 is not a purely specific marker for Tregs in humans, because it is also expressed by some conventional Th cells, as acknowledged by Shimon Sakaguchi, the father of Tregs (PMID: 19464196). Actually, most human CD4 and CD8 T cells seem to upregulate FOXP3 upon activation (e.g.: PMID: 17329235, 17154262). The authors need to make this point very clear in the manuscript and discuss the implications for the interpretations of their results.
- 4) Materials and Methods, page 11. The "minimum P-value approach" that was used should be briefly described rather than only referring to previous publications.
- 5) The authors claim that FOXP3+ Tregs in TLS express CD62L and apparently used CD62L as a marker for TLS-Tregs throughout the manuscript (this point needs to be explained better). However, the CD62L staining shown in Figure 1H is not convincing. The staining is weak and seem to be present outside the TLS as well! The authors need to present more convincing evidence that TLS-Tregs are indeed CD62L positive. They should also discuss the possibility that not all TLS-Tregs may be CD62L positive.
- 6) Figure 1I. For clarity, the y-axis could be re-labelled "CD62L-/CD62L+ T cells"
- 7) Figure 1J. The complete gating strategy should be shown. It is unclear how the two dot plots on the right-hand side relate to the rest. Isotype-matched control mAb staining should be shown for CD62L.
- 8) Figure 3A. Gene expression data suggest that "tumor Tregs" express the canonical Th1 cytokines TNF and IFN-gamma, which is very surprising (actually, it does not make sense). How do the authors explain this? Can they exclude a Th1 cell contamination?
- 9) Page 15. The conclusion "Altogether, Tregs have a distinct activation and immunosuppressive pattern" needs to be moderated if the authors mean that "tumor Tregs" produce the inflammatory cytokines TNF and IFN-gamma (see previous comment).
- 10) Figure 6 and page 17. Anti-GITR and anti-CTLA-4 antibodies were used to investigate the molecular mechanism of suppression by Tregs. However, both molecules are immune checkpoints, and their blockade may affect both Tregs and Tconv. The authors need to mention this possibility and its implication for the interpretation of their data.
- 11) Figure 7. What are the relationships between the immunological data, patient prognosis and NSCLC disease stages? Do the immunological data represent prognostic indicators independently of cancer stage? Please clarify.
- 12) Supplementary Figure 5. Were TLS-Tregs quantified by immunostaining or flow cytometry. Please clarify.
- 13) How informative are TLS-Tregs for patient prognosis? The p-value in Supplementary Figure 5A is much higher than in Figure 7A. This point should be mentioned and discussed in the manuscript.
- 14) Discussion, page 19. "We observed that CD4+CD25hiCD127-FoxP3+ Tregs are prevalently distributed in the stroma and tumor-induced TLS but are rarely found in direct contact with tumor cells." This conclusion is certainly not supported by the data presented because it would require quadruple immunostaining of tumor sections ! Please correct.
- 15) Discussion, page 23. "In summary, the high density of Tregs in whole tumor and especially in TLS is associated with the reduced survival of NSCLC patients. " What is the basis for the "especially in TLS"? See comment #13 above. Please correct this statement.

Reviewer #2 (Remarks to the Author):

The topic of TLS composition and manipulation of Treg function for the benefit of cancer patients has high translational potential and is of high interest to general audience.

The manuscript suffers from 2 major weaknesses:

1)lack of novelty, which is a pre-requisite for publication in Nature Communications Biology. The suppressive effect of Tregs found in tumours is well known (e.g. PMID: 31554638) and the adverse correlation of Tregs presence and clinical outcome in different cancer entities, including lung cancer, has been already established (review: 2015 Zhao et al, Oncotarget, PMID: 27153545). The TLS - association of Tregs is interesting, however, the described phenomenon was already published for breast cancer (Gobert et al, Cancer Research 2009, PMID: 19244125). Findings presented in Figure 4 on the elevated expression of co-stimulatory receptors on Tregs wrt non-Tregs are also known. Therefore, submitted manuscript only extends some of the findings to lung cancer entity. There is no novel mechanistic insight into role of Tregs in the TLS, and no novel therapeutic avenue was suggested that can be exploited after presented study.

2)flawed experimental design. The results of Treg gene expression profiling based on their location within the TLS is potentially flawed. Crucially, it is not clear why authors discriminate between Tregs present within the TLS vs Tregs outside the TLS based on CD62L expression by Tregs. This is not accepted by the scientific community; therefore this approach needs to be first vigorously proven to be accepted for further research and any results generated downstream of this finding. Actually, the fact that the gene expression profile of Tregs found inside and outside of the TLS is very similar, can be interpreted as an artifact of the false basis on discrimination of two subsets based on CD62L+ expression. Generated dataset on gene expression within Treg populations is not novel and similar data is already publicly available from higher resolution sequencing technologies (such as single cell seq).

Dear Reviewers,

Please find enclosed a point-by-point reply.

Yours sincerely,

Marie-Caroline Dieu-Nosjean

Reviewer #1:

First, we would like to thank Reviewer 1 for his/her comments and suggestions to greatly improve the quality of our manuscript. Please find below how we updated the text and figures, accordingly.

1) The text of the manuscript needs some serious improvements to improve its clarity. The current text is difficult to follow. The authors need to explain better what exactly they have done and found.

R1: we thank the Reviewer for this recommendation. The manuscript, figures, tables, and legends have been revised according to the points raised below.

2) Several reports have previously shown an association between high numbers of Tregs in NSCLC tumors and worse prognosis (e.g.: PMID: 20234320, 21719142, 23269987). The authors need to clearly refer to this previous work in the Introduction and compare their new findings with these previous studies in the Discussion.

R2: Indeed, we chose to talk about the prognostic value of Treg in most solid tumors in order to have a global overview. Four articles out of the 50 references cited in our manuscript deal with this topic, including Gentles et al. (Nature Med, 2015), who show that densities of Treg are not related to the clinical outcome of cancer patients (i.e. 25 human cancer types, 5,782 tumors) but tend to be associated with a favorable prognosis in lung adenocarcinoma (Fig. 3d). The latter point is contradictory to the 3 articles suggested by Reviewer 1.

In addition, 2 out of the 3 articles suggested by Reviewer 1 use a single FoxP3 staining for the quantification of Treg on tumor sections (PMID20234320=Shimizu et al., J Thorac Oncol, 2010; PMID21719142=Tao et al., Lung Cancer 2011). Now, we are sure that FoxP3 is not enough to affirm that stained cells are Treg. Indeed, other cell types such as granulocytes, macrophages, and alveolar cells type 2 can express FoxP3 (<https://www.proteinatlas.org/ENSG00000049768-FOXP3/single+cell+type>). Like Reviewer 1, we read with great interest the publication of Suzuki and colleagues (PMID: 23269987). The authors focused the study to stage I lung adenocarcinoma which represents only a minority of our cohort of NSCLC patients (stages I to IV, adenocarcinoma and squamous cell carcinoma).

Altogether, the prognostic value of tumor-infiltrating Treg is still a matter of debate, and no obvious association has been clearly depicted with poor survival in cancers, including lung cancer, and this explains why we decided to be very cautious.

3) FOXP3 is not a purely specific marker for Tregs in humans, because it is also expressed by some conventional Th cells, as acknowledged by Shimon Sakaguchi, the father of Tregs (PMID: 19464196). Actually, most human CD4

and CD8 T cells seem to upregulate FOXP3 upon activation (e.g.: PMID: 17329235, 17154262). The authors need to make this point very clear in the manuscript and discuss the implications for the interpretations of their results.

R3: We totally agree with Reviewer 1, and we have taken this into careful consideration in our analysis. This is the reason why we deeply investigated the phenotype of FoxP3 positive cells by flow cytometry. We defined Tregs as CD3+ CD4+ FoxP3bright CD25bright CD127- cells (now Fig. 2c) in accordance with the literature in human studies (Fehervari et al., JCI, 2004; Fontenot et al., Nature Immunol., 2005 ; Liu et al., JEM, 2006) (page 14, lines 260-262). Moreover, the transcriptomic analysis of freshly isolated CD2+ CD8- CD4+ CD25bright CD127- cells confirmed that they express the highest level of Treg markers i.e. FoxP3, IL-2RA, TIGIT, TNFR2, GITR, and CTLA-4 (now Fig. 5a-b). Thus, it allows us to use both CD3 and FoxP3 antibodies for the quantification of CD3+ FoxP3++ Tregs by IHC on FFPE tumor sections. Finally, we never observed any CD8+ FoxP3+ Tregs in tumors of NSCLC patients (page 14, lines 260- 261).

The phenotype of Tregs by flow cytometry was also clearly mentioned in the discussion of the revised version (page 21, lines 462-464).

4) Materials and Methods, page 11. The "minimum P-value approach" that was used should be briefly described rather than only referring to previous publications.

R4: Thank you for this remark. We better described this step in the revised version of the manuscript (page 12, lines 230-231).

5) The authors claim that FOXP3+ Tregs in TLS express CD62L and apparently used CD62L as a marker for TLS-Tregs throughout the manuscript (this point needs to be explained better). However, the CD62L staining shown in Figure 1H is not convincing. The staining is weak and seem to be present outside the TLS as well! The authors need to present more convincing evidence that TLS-Tregs are indeed CD62L positive. They should also discuss the possibility that not all TLS-Tregs may be CD62L positive.

R5: We previously published that TLS-T cells selectively express CD62L in contrast to non-TLS-T cells in NSCLC patients (de Chaisemartin et al., Cancer Res, 2011, Fig. 1F-G). Based on Reviewer's request, we specifically set-up a new multi-IF staining on FFPE tumor sections. Of note, we were waiting for more than 4 months for the shipment of the anti-CD62L antibody for the IHC/IF application on FFPE samples. Based on the CD20/CD3/FoxP3/CD62L/DAPI multi-IF staining, the TLS areas were defined by the presence of a CD20+ B cell zone adjacent to a CD3+ T cell zone. We confirmed our previous observation that CD62L expression is restricted to TLS-T cells including Tregs. Indeed, CD3+ FoxP3bright Treg selectively express CD62L in TLS whereas Tregs in other areas of the tumor do not (updated Fig. 1g). Thus, panels (g) and (h) of Figure 1 were replaced by the new multi-panels (g) with a focus on TLS (upper line) and non-TLS areas (bottom line). We hope that the Reviewer will appreciate the technical challenge that of developing a new 5plex-IF panel on FFPE samples in a very short period of time.

6) Figure 1I. For clarity, the y-axis could be re-labelled "CD62L-/CD62L+ T cells"

R6: The y-axis of the previous Fig. 1I (now Fig. 2a) was modified accordingly.

7) Figure 1J. The complete gating strategy should be shown. It is unclear how

the two dot plots on the right-hand side relate to the rest. Isotype-matched control mAb staining should be shown for CD62L.

R7: The first steps of the gating strategy are shown in supplementary Fig. 1a (from FSC/SCC to CD25/FoxP3 dot-plot). The right and left parts of the previous Fig. 1j (now Fig. 2b-c) come from two independent experiments. In order to avoid any confusion by readers, we split these data into 2 distinct panels (b and c). We also added the dot-plot isotype-matched control for CD62L in the panel b (new Fig. 2b) with the percentage of cells in each gate, according to the Reviewer's request.

8) Figure 3A. Gene expression data suggest that "tumor Tregs" express the canonical Th1 cytokines TNF and IFN-gamma, which is very surprising (actually, it does not make sense). How do the authors explain this? Can they exclude a Th1 cell contamination?

R8: We understand very well the caution made by Reviewer 1, and we also checked this point as soon as we started the transcriptomic analyses. First, the purity of the cell sorting was between 98 to 100%. Secondly, the volcano-plot of the previous Fig. 3a (new Fig. 4a) simply shows that TNF-a and IFN-g mRNA are significantly overexpressed on tumor-infiltrating Tregs compared to blood Tregs. Even statistically significant, the expression level of these two genes remains quite low in NSCLC patients (mean of TNF-a mRNA: 115 and 364 counts for blood Tregs and TIL-Tregs, respectively; mean of IFN-g mRNA: 76 and 432 counts for blood Tregs and TIL-Tregs, respectively).

Finally, Vignali *et al.* nicely showed that a high proportion of intra-tumoral Tregs produce IFN-g in a mouse model of melanoma (Overacre-Delgoffe et al., Cell, 2017). Thus, it is not formally excluded that this cytokine can be expressed at low level by Tregs in some human tumors.

9) Page 15. The conclusion "Altogether, Tregs have a distinct activation and immunosuppressive pattern" needs to be moderated if the authors mean that "tumor Tregs" produce the inflammatory cytokines TNF and IFN-gamma (see previous comment).

R9: We agree that this sentence is misleading. We wanted to emphasize that the balance of activator/immunosuppressive receptors is distinct on Tregs isolated from tumors *versus* the other anatomical sites. We changed the sentence as follows: "Altogether, Tregs have a distinct expression pattern of activation and inhibitory receptors in tumors compared with LN and blood" (page 16, lines 325-326).

10) Figure 6 and page 17. Anti-GITR and anti-CTLA-4 antibodies were used to investigate the molecular mechanism of suppression by Tregs. However, both molecules are immune checkpoints, and their blockade may affect both Tregs and Tconv. The authors need to mention this possibility and its implication for the interpretation of their data.

R10: The functional studies were focused on ICP such as GITR and CTLA-4 proteins that were highly expressed on tumor-infiltrating Tregs compared to CD4+ Tconv (data shown in previous Fig. 4d/now Fig. 5d). Thus, the aim of the *ex vivo* experiments was to assess the impact of ICP blockade targeting Tregs on the proliferation of CD4+ Tconv. As mentioned by Reviewer 1, we also evaluated the putative direct effect of neutralizing antibodies on CD4+ Tconv. In the previous Fig. 6a (now Fig. 7a), the first 3 panels from the left show CD4+ Tconv cultivated alone (without any Tregs) in presence of anti-GITR or anti-CTLA-4 antibody. No difference

was detected compared to the isotype control (55.4% of proliferation with the isotype control versus 56.1% with anti-GITR, and 55.6% with anti-CTLA-4). Thus, we can conclude that the neutralizing antibodies against GITR and against CTLA-4 had no direct effect on the proliferation of CD4⁺ Tconv. We changed the paragraph in the revised manuscript “Anti-GITR and anti-CTLA-4 antibodies had no direct impact on the proliferation of CD4⁺ Tconv cultivated alone, indicating that these antibodies have no direct effect on this population. However, the proliferation of TIL-CD4⁺ Tconv was totally recovered when co-cultivated with Tregs and antibodies against GITR and CTLA- 4” (page 18 lines 377-381).

11) Figure 7. What are the relationships between the immunological data, patient prognosis and NSCLC disease stages? Do the immunological data represent prognostic indicators independently of cancer stage? Please clarify.

R11: We thank the Reviewer for this excellent question. We performed several statistical analyses, accordingly. A multivariate Cox proportional hazard (CPH) model (Cox et al., 1972) was implemented in order to estimate relationships between the immunological data, patient prognosis and NSCLC disease stages. NSCLC disease stages IIIB (n=6) et IV (n=2) were grouped into a single category. Prior variable selection was performed using univariate modelling. Sensitivity analyses included alternatives for eventual violation of the proportional hazard assumption (PHA, tested via Grambsch & Therneau approach, 1994) and a Cox Lasso approach (Tibshirani, 1997) for the variable selection, using 10-fold cross-validation to determine hyperparameter value and to study impact on partial likelihood deviance and coefficients. Methods are described in the Material and Methods, accordingly (pages 12-13, lines 232-239).

Briefly, both immunological data and NSCLC disease stages were significantly associated with patient prognosis in unadjusted and adjusted CPH models (new Table 1). Furthermore, as NSCLC disease stages and immunological data were significantly associated one to another at 5% threshold (Chi-square test p-value = 0.042), NSCLC disease stage was deemed as a confounding factor in adjusted CPH model. Significant bad prognosis factors were stages IIIA (HR (95% CI) = 3.15 (1.45 – 6.84) compared to stage IA), Treg^{Hi} TLS-DC^{Lo}/CD8^{Lo} (HR (95% CI) = 4.00 (1.08 - 14.84) compared to 3 Hi) and interaction between stage IIA and 3 Lo (HR (95% CI) = 13.53 (1.75 – 104.50) compared to stage IA 3 Hi). A forest plot of adjusted hazard ratios is available in Supplementary Figure 6. Sensitivity analysis with Cox Lasso approach led to identical variable selection (pages 20-21, lines 434-446).

12) Supplementary Figure 5. Were TLS-Tregs quantified by immunostaining or flow cytometry. Please clarify.

R12: We apologize for this confusion. TLS-Tregs were quantified by immunostaining. We updated the legend of supplementary Fig. 5 as followed “Densities of total Tregs, TLS-mature DC, and CD8⁺ T cells were determined by IHC using serial sections of FFPE lung tumor samples” (page 38, lines 888-889).

13) How informative are TLS-Tregs for patient prognosis? The p-value in Supplementary Figure 5A is much higher than in Figure 7A. This point should be mentioned and discussed in the manuscript.

R13: previous Fig. 4c (now Fig 5c) shows a selective molecular pattern of Tregs in TLS versus non-TLS suggesting that they may exhibit distinct immune function in

these two areas. Thus, this was the reason why we investigated the prognostic value of Tregs in the context of TLS. This question was not trivial as TLS-Tregs express higher level of IL-6 compared to non TLS-Tregs. IL-6 is a cytokine involved in the differentiation of B-cells and activation of follicular-helper T cells, two key players of the immune function of TLS. We demonstrated for the first time that TLS-Tregs correlate with a worse prognosis in primary NSCLC, as observed for non TLS-Tregs. Thus, high Treg infiltrate whatever their location (i.e. in TLS or not), significantly correlates with short-term survival of NSCLC patients.

In statistics, the p-value is the probability of obtaining results at least as extreme as the observed results of a statistical hypothesis test, assuming that the null hypothesis is correct. A smaller p-value means that there is stronger evidence in favor of the alternative hypothesis. Even if the p-value is higher for TLS-Tregs compared to total Tregs (0.0245 vs 0.0040, respectively), it is wrong to affirm that the prognostic value of TLS-Treg is lower than total Tregs. The p-value is significant for total Tregs and TLS-Tregs, indicating that each variable is related to patient survival (page 25, lines 535-537).

14) Discussion, page 19. “We observed that CD4⁺CD25^{hi}CD127⁻FoxP3⁺ Tregs are prevalently distributed in the stroma and tumor-induced TLS but are rarely found in direct contact with tumor cells.” This conclusion is certainly not supported by the data presented because it would require quadruple immunostaining of tumor sections ! Please correct.

R14: Sorry for this mistake. The experiment was performed by IHC on tumor sections. We changed the sentence to “We observed that CD3⁺ FoxP3⁺ Tregs are prevalently distributed in the stroma and tumor-induced TLS but are rarely found in direct contact with tumor cells” (page 21, lines 461-462).

15) Discussion, page 23. “In summary, the high density of Tregs in whole tumor and especially in TLS is associated with the reduced survival of NSCLC patients. “ What is the basis for the “especially in TLS”? See comment #13 above. Please correct this statement.

R15: We apologize for this confusion. We replaced “especially” by “including” (page 25, lines 545-546).

Reviewer #2 (Remarks to the Author):

The topic of TLS composition and manipulation of Treg function for the benefit of cancer patients has high translational potential and is of high interest to general audience.

The manuscript suffers from 2 major weaknesses:

1) Lack of novelty, which is a pre-requisite for publication in Nature Communications Biology. The suppressive effect of Tregs found in tumours is well known (e.g. PMID: 31554638) and the adverse correlation of Tregs presence and clinical outcome in different cancer entities, including lung cancer, has been already established (review: 2015 Zhao et al, Oncotarget, PMID: 27153545).

R1.1: In the introduction, we summarized some studies dealing with the prognostic value of Tregs in most solid tumors in order to have a global overview. Among the

four cited articles, we mentioned Gentles et al. (Nature Med, 2015) who show that densities of Tregs are not related to the clinical outcome of cancer patients (i.e. 25 human cancer types, 5,782 tumors) but tend to be associated with a favorable prognosis in lung adenocarcinoma (Fig. 3d), the latter point is contradictory to some publications among with Zhao et al. (PMID: 27153545, 2016).

Thus, the prognostic value of tumor-infiltrating Treg is still a matter of debate even in lung cancer, and we postulated that the composition of the immune microenvironment may, in part, explain this apparent discrepancy. So, we decided to investigate this point in the context of TLS, a key player for the next generation of immunotherapy (Petitprez et al., Nature 2020; Cabrita et al., Nature 2020, Helmink et al., Nature 2020). For the first time, we demonstrated that: 1- the density of TLS-Tregs is associated with a poor clinical outcome, and 2- the combination of Tregs with TLS-DC and CD8+ T cells allows the best stratification of NSCLC patients compared to each variable alone (patent world wide: 2017/032867). None publication has compared the transcriptomic signature of non-TLS-Tregs versus TLS-Tregs. We are the first study highlighting selective gene expression by each Tregs subset according to their *in situ* localization in the human lung tumors (n=20 NSCLC patients).

The TLS -association of Tregs is interesting, however, the described phenomenon was already published for breast cancer (Gobert et al, Cancer Research 2009, PMID: 19244125).

R1.2: I discussed several times with the excellent Team of C. Caux, and they never claimed that Tregs home to TLS in BC patients as they were not convinced themselves. It is the reason why the authors mentioned “lymphoid infiltrates” (TLS is never mentioned) “ in Gobert’s article.

Findings presented in Figure 4 on the elevated expression of co-stimulatory receptors on Tregs wrt non-Tregs are also known. Therefore, submitted manuscript only extends some of the findings to lung cancer entity. There is no novel mechanistic insight into role of Tregs in the TLS, and no novel therapeutic avenue was suggested that can be exploited after presented study.

R1: We agree with Reviewer 2 that the ICP profile of Tregs and non-Tregs has been published in several solid tumors. The comparison of the transcriptomic analyses was not the goal of our study but rather a control of our cohort. Thus, CD4+ Tconv were used as a reference. Again, the novelty of our study is to investigate Tregs in the context of TLS. Thus, we compared the transcriptome of Treg versus CD4+ Tconv in TLS, same in non-TLS areas, and finally TLS-Tregs versus non-TLS Tregs, an investigation that has never been published before thanks to CD62L marker (see also response to Q2 below). This question was not trivial as TLS-Tregs express higher level of IL-6 compared to non TLS-Tregs. Il-6 is a cytokine involved in the differentiation of B-cells and activation of follicular-helper T cells, two key players of the immune function of TLS. Our study is the first one reporting the transcriptomic profile of TLS-Tregs. We would have liked to investigate a mechanistic insight into the role of TLS-Tregs but, unfortunately, their very low number in tumors did not allow us any further investigation (i.e. functional studies ex vivo).

Finally, regarding the conflicting view of the mechanism of CTLA-4 blockade, here we show evidence that this strategy might allow the recovery of CD4+ Tconv function, same for GITR but others such as ICOS, PD1, TIGIT or Tim-3 may not impact the immunosuppression mediated by TIL-Tregs on Tconv in NSCLC. These data are of

high interest to clinicians since many ongoing clinical trials target these ICP. In line with Reviewer's comment, many other targets have to be tested but the COVID pandemic makes any further experiments from fresh samples of NSCLC patients very difficult. However, our current data showing a differential transcriptomic signature between TLS-Tregs and non-TLS-Tregs may have some critical consequences regarding ICP blockade in patients (ICP targeting both Treg subsets such as anti-CTLA-4, or mainly one such as anti-PD1, anti-GITR for non-TLS-Tregs) (page 24, lines 526-528).

2)flawed experimental design. The results of Treg gene expression profiling based on their location within the TLS is potentially flawed. Crucially, it is not clear why authors discriminate between Tregs present within the TLS vs Tregs outside the TLS based on CD62L expression by Tregs. This is not accepted by the scientific community; therefore this approach needs to be first vigorously proven to be accepted for further research and any results generated downstream of this finding.

R2.1: We totally understand the Reviewer's comment as CD62L was a critical marker in the study. We previously published that TLS-T cells selectively express CD62L in contrast to non-TLS-T cells in NSCLC patients (de Chaisemartin et al., Cancer Res, 2011, Fig. 1F-G). Based on Reviewer's request, we specifically set-up a new multi-IF staining on FFPE tumor sections. Of note, we were waiting for more than 4 months for the shipment of the anti-CD62L antibody for the IHC/IF application on FFPE samples. Based on the CD20/CD3/FoxP3/CD62L/DAPI 5plex-IF staining, the TLS areas was defined by the presence of CD20+ B cell zone adjacent to a CD3+ T cell zone. We confirmed our previous observation that CD62L expression is restricted to TLS-T cells including Tregs. Indeed, CD3+ FoxP3bright Treg selectively express CD62L in TLS whereas Tregs in other areas of the tumor do not (updated Fig. 1g). Thus, panels (g) and (h) of Figure 1 were replaced by the new multi-panels (g) with a focus on TLS (upper line) and non-TLS areas (bottom line). We hope that the Reviewer will appreciate the immense technical challenge involved in developing a new 5plex-IF panel on FFPE samples in a very short period of time.

Actually, the fact that the gene expression profile of Tregs found inside and outside of the TLS is very similar, can be interpreted as an artifact of the false basis on discrimination of two subsets based on CD62L+ expression. Generated dataset on gene expression within Treg populations is not novel and similar data is already publicly available from higher resolution sequencing technologies (such as single cell seq).

R2.2: We also are very interested in this critical point, and we thank the Reviewer for asking this question. As mentioned previously, new multiplex-IF staining on NSCLC sections (Fig. 1g) demonstrates that CD62L is a selective marker for TLS-T cells (as previously reported by de Chaisemartin et al., Cancer Res, 2011) including TLS-Tregs (current data). This observation is in accordance with the literature showing that the adhesion molecule CD62L is a lymphoid homing molecule, and CD62L/PNAd may play a similar role in lymphocyte HEV-mediated recruitment in cancer-associated TLS (see also Martinet et al., Cancer Res, 2011).

Even if few genes are differentially expressed between TLS-Tregs and non-TLS Tregs, CCR7 is overexpression by Tregs in TLS, a zone where both CCR7 ligands, CCL19 and CCL21 are highly secreted (de Chaisemartin et al., Cancer Res, 2011). Concerning non-TLS Tregs, they overexpressed many ICP suggesting a strong

immunosuppressive activity (the sentence was modified, accordingly; page 18 lines 362-364).

We totally agree that scRNASeq will be more informative. However, we have to keep in mind that the number of TLS-Tregs is extremely low compared to non-TLS Tregs. And without any positive selection of TLS-Tregs, the analysis of this subset will be compromised. It is the reason why we did not analyze public database. We decided to sort TLS-Tregs and non-TLS Tregs for such study.

Reviewers' comments:

Reviewer #1 (Remarks to the Author):

The manuscript has been considerably improved. However, a number of remaining minor issues need to be addressed before publication.

Specific comments:

1. Title. "Regulatory T cells ... impose poor clinical outcome" is a too strong conclusion that is not supported by the data because a causal link between the presence of Tregs in tumor and worse prognosis has not been shown. "Regulatory T cells ... are associated with poor clinical outcome" would be more accurate.
2. Abstract. "Tregs in the whole tumor, and particularly in TLS, are associated with a poor outcome". What do the authors mean with "particularly"? Please clarify.
3. Abstract. "Targeting Tregs especially in TLS may represent a major issue". What do the authors mean by "issue"? An approach? (Definition of issue: "a subject or problem that people are thinking and talking about").
4. The presence of a high number of Tregs (and Th2 cells) in NSCLC tumors and TLS has been recently reported by Frafjord et al (PMID: 34868011). This relevant paper should be mentioned and referred to in the Introduction or Discussion of the manuscript.
5. Figure 8. Please specify in the figure legend how Tregs were identified in tumor sections. Were only CD3+Foxp3+ double positive cells counted?
6. Page 19 and Figure 8. What is the definition of "TIL-Tregs" on tumor sections? Were TLS included or excluded for the counting of TIL-Tregs? This important point should be clearly explained in the text and figure legend.
7. Supplementary Figure 5. These graphs are essential for the conclusions of the manuscript (and are even mentioned in the title!) and should be presented as a main figure instead (Figure 9). An additional survival graph with only TLS-DC/CD8hi, TLS-DC/CD8mix, and TLS-DC/CD8lo (without Treg data) should be shown in the same figure for comparison.

Reviewer #2 (Remarks to the Author):

I am impressed by authors efforts to clarify and rebut many significant points raised in a review of their paper. The manuscript has clearly improved in quality, it is much easier to recognise and appreciate the novelty brought by the study.

The authors have provided significant evidence of the specificity of the CD62L staining as a TLS marker, in the form of newly established multiplex IF staining. This strengthened validity of major findings from the transcriptome analysis. The potential for re-invigoration of the Tregs (associated with the TLS) by anti-CTLA4 and anti-GITR antibodies, but not other immune cell receptors, is an important piece of information in light of clinical efforts in the field of cancer immunotherapy.

I approve the manuscript for the publication in the current version.

Reviewer #3 (Remarks to the Author):

Devi-Marulkar et al. show that Tregs have similar immune profiles in TLS (lung tumors) and non-TLS (distant normal tissue) areas of the tissues examined. They further show that the Treg TIL are functional and their suppressive abilities can be abrogated by blocking the immune checkpoint molecules CTLA-4 and GITR. They find that Tregs in the tumor, but particularly in the TLS, are associated with poor clinical outcome. And TLS Tregs together with mature DC or CD8 T cells

distinguish lung cancer patients with the highest risk of death.

Comments on the original manuscript:

The data presented in this manuscript are interesting but due to their primarily descriptive nature they do not greatly advance our understanding of the interplay between Treg and conventional T effector cells in the tumor microenvironment and TLS. The group of tumors analyzed are extremely heterogeneous in the balance between Treg and conventional CD4 T cells. The authors could have reinforced their conclusions by taking these descriptive observations one step further to link the balance of Treg to Tconv in a specific region (ex TLS or non-TLS, B cell follicle or T cell zone of a TLS, etc) and between tumors with more/less TIL and or TLS and survival.

This study examines many facets of TLS in the tumor but does not provide insight into what is happening in the balance between conventional T cell activation and Treg-mediated suppression that leads to better or worse survival.

Specific questions:

1. The similarities between Tregs in tumors and non-tumor distant lung suggests that these tissue localized cells have differentiated and matured similarly before entering the tissue. What about other tissues outside of the lung – are they similar or different? Do they also show this activated, effector-memory phenotype? Is this just a reflection of their ability to enter tissue locations and as such is not specifically influenced by the tumor microenvironment?
2. Is there a difference in activity between TLS with higher levels of Tregs compared to those with lower levels? In other words, are these TLS less active in other ways?
3. Is the sorted population of Tregs variable in the number of suppressive relative to activated T cells? Did you check Foxp3 expression (protein by flow cytometry) and gene status (demethylation)? Differences in the ratio of activated to suppressive CD2+CD4+CD8-CD25hiCD127- T cells could alter the activity of the TLS.
4. What exactly are the areas defined as non-TLS? Are they aggregates where the CD4, CD8 and B cells are randomly arranged or are they areas where there are congregations of T cells or B cells that could be considered small groups?
5. In the TLS, are the Tregs more localized in the B cell follicle or in the T cell zone?
6. Is the heterogeneity of Tregs relative to conventional T cells a reflection of their overall ability to suppress the activities of the latter? Or are there other factors in the tumor that contribute to immunosuppression? It is interesting that there is similar heterogeneity in the LN although the ratios are lower in the LN. Can you comment on their known activities in LN's?
7. Is there any evidence that the Treg are differentiated from conventional T cells in the tissues compared to being recruited after differentiation?
8. Are the differences in gene expression comparing Tregs from blood with tumors relevant? The cells in the blood are known to not be active. Because there were less pronounced differences in the comparison with LN and distant normal tissues, maybe the differences with the blood are just alterations associated with infiltrating tissues in general and not specific to the tumor?
9. The molecular profiles of Tregs (compared to conventional CD4 T cells) in and outside of the TLS are interesting but highly descriptive. It would be nice to see some functional link between expression data and activity. Are there any differences in surface receptors (like CXCR3 or others that differ in

staining intensity) that could be used to sort and test them separately in the functional assay?

10. What does it mean that only blocking GITR or CTLA-4 but not ICOS, PD-1, TIGIT or TIM-3 reversed Treg-mediated suppression of conventional T cells. If suppression is TGFb or IL-10 mediated then is there a link?

11. What is the difference in a TLS with low Tregs, high DC and high CD8 compared to a high Treg, low DC, low CD8 TLS? Are there other characteristics that vary between these two types of TLS that could also reflect survival?

12. The speculation in the discussion should be removed.

Comments on the revised manuscript:

While I found the manuscript greatly improved in response to Reviewer 1 & 2's comments there are still some open questions.

1. The authors argue that the presence of regulatory T cells (high versus low) in the TLS is associated with a poor prognosis. As an extension of the comment on Treg/Tconv balance mentioned in the original review comments (underlined), this subject has not been addressed in the revised manuscript. There are multiple studies of viral infection, autoimmune disease and more recently cancer, showing that the balance of T effector cells to T regulatory cells is an important determinant of activation or suppression. How do the authors explain this? Can they calculate the balance between Treg and Tconv in their samples and correlate this with response? This would be extremely informative.

2. The authors used a customized array for the gene expression study and limited markers in flow cytometric analysis. It is unfortunate that they did not include other important markers of TLS such as CXCR5 [this could have been used in flow in combination with CD62L to show whether the TLS-associated Treg were in fact all or partially Tfr (T follicular regulatory cells)]. Multiple studies have shown that TLS contain not only CXCR5+ B cells but also CXCR5+ T cells, both CD4 and CD8.

3. It is curious that the authors find Tconv are mostly naïve and central memory T cells in TLS – many studies show that they are more mature (EM1-EM4) memory cells. Can they comment on this?

4. Is the higher expression of PD-1 on conventional T cells compared to Treg due to a greater follicular helper T cell presence?

5. Discussion p22, lines 425-8: There is no demonstration that CCR4 and the ligands CCL17 and CCL22 are expressed as protein in Tregs so this statement is speculation without even a reference to back it up (FACS, IF, or the chemokines in the SN).

6. Discussion p24, lines 485-8 is again speculation based on gene expression analysis and needs a demonstration that CXCR3 protein is expressed on the surface of these cells.

Point-by-point response to the Reviewers

A point-by-point reply to the Reviewer's comments is found below. The reviewers's comments are indicated in blue, followed by our responses in black.

Reviewer #1 (Remarks to the Author):

The manuscript has been considerably improved. However, a number of remaining minor issues need to be addressed before publication.

Specific comments:

1. Title. "Regulatory T cells ... impose poor clinical outcome" is a too strong conclusion that is not supported by the data because a causal link between the presence of Tregs in tumor and worse prognosis has not been shown. "Regulatory T cells ... are associated with poor clinical outcome" would be more accurate.

MCD: We changed the title accordingly as followed "Regulatory T cells infiltrate the tumor-induced tertiary lymphoid structures and are associated with poor clinical outcome of patients with lung cancer". Of note, none of the Reviewers requested to change it following the reviewing of the first version of the manuscript.

2. Abstract. "Tregs in the whole tumor, and particularly in TLS, are associated with a poor outcome". What do the authors mean with "particularly"? Please clarify.

MCD: To clarify the sentence, we substituted « particularly » for « including ». It means that total Tregs (whole tumor) and TLS-Tregs (Tregs present in TLS, determined among total Tregs) are associated with poor clinical outcome in NSCLC. The new sentence is "Tregs in the whole tumor, including in TLS, are associated with a poor outcome of NSCLC patients"

3. Abstract. "Targeting Tregs especially in TLS may represent a major issue". What do the authors mean by "issue"? An approach? (Definition of issue: "a subject or problem that people are thinking and talking about").

MCD: We thank the Reviewer for highlighting this mistake. We changed "issue" by « major challenge ». The new sentence is "Thus, Targeting Tregs especially in TLS may represent a major challenge in order to boost anti-tumor immune responses initiated in TLS".

4. The presence of a high number of Tregs (and Th2 cells) in NSCLC tumors and TLS has been recently reported by Frafjord et al (PMID: 34868011). This relevant paper should be mentioned and referred to in the Introduction or Discussion of the manuscript.

MCD: Frafjord and colleagues (Frontiers Immunol, 2021) quantified the five main Th cell subsets (Th1, Th2, Th17, Tfh and Treg) in 11 NSCLC by 4-colour multiplex IHC. The aim of their study is very interesting but some limitations have to be raised. Th cells were identified as CD3⁺ CD8⁻ cells. However, we and others have observed by flow cytometry that CD3⁺ CD8⁻ T cells also include numerous double negative CD4⁻

CD8⁻ cells which do not belong to CD4⁺ Th compartment (see below data from Supplemental Fig. 1, Goc et al., Cancer Res, 2014).

Moreover, the predominance of Tregs (and Th2) *versus* Th1 (cells per mm²) reported by Frafjord et al. was only statistically significant in the tumor stroma (no difference in the tumor epithelium) and in adenocarcinoma (no difference in squamous cell carcinoma) (Fig. 3A-D). Of note, this study was done in 6 adenocarcinomas which is low regarding the heterogeneity of the tumor microenvironment in NSCLC.

Finally, we have previously published in a prospective cohort of 28 NSCLC patients that the presence of DC-Lamp⁺ mature DC (hallmark of TLS, as done in the present study) correlates with a specific intra-tumoral immune contexture characterized by the overexpression and the coordination of genes related to Th1 orientation, T cell activation and cytotoxic effector functions (Fig. 2, Goc et al., Cancer Res., 2014). No correlation was observed between TLS and Th2 signature or immunosuppression. One explanation between these apparent discrepancies could be the limited number of NSCLC patients included in the Frafjord's study.

Thus, as requested by the Reviewer #1, we referred the Frafjord's study but we indicated in the Discussion section : « Frafjord et al. reported the predominance of Tregs (and Th2) *versus* Th1 (cells per mm²) (Frontiers Immunol, 2021) while we have previously reported that the TLS presence correlates with a specific intra-tumoral immune contexture characterized by the overexpression and the coordination of genes related to Th1 orientation, T cell activation and cytotoxic effector functions (Goc et al., Cancer Res., 2014). By contrast, no correlation was observed between TLS and Th2 signature or immunosuppression (Goc et al., Cancer Res., 2014). These apparent discrepancies could be due to the limited number of NSCLC patients included in the Frafjord's study » (page 25, lines 512-518).

5. Figure 8. Please specify in the figure legend how Tregs were identified in tumor sections. Were only CD3⁺Foxp3⁺ double positive cells counted?

MCD: We modified the sentence of the figure legend 8 (page 35, lines 794-796) as followed: "Densities of total CD3⁺ FoxP3⁺ Tregs (i.e. TIL-Tregs infiltrating the whole tumor section), TLS-DC-Lamp⁺ mature DC, and CD8⁺ T cells were determined using serial sections of tissues (n=338 NSCLC)".

6. Page 19 and Figure 8. What is the definition of “TIL-Tregs” on tumor sections? Were TLS included or excluded for the counting of TIL-Tregs? This important point should be clearly explained in the text and figure legend.

MCD: “TIL-Tregs” means total Tregs infiltrating the tumor, as defined previously (TIL: Tumor-infiltrating lymphocytes). Here, TIL-Tregs were quantified on the whole tumor section based on the CD3/FoxP3 double staining performed on FFPE tumor section. Tumor areas includes all sub-areas of the tumor, i.e., tumor nests, TLS and other areas of the stroma. The methods used for the cell quantification are explained in the Online methods supplement/Cell quantification section. “CD3⁺ total T cells and CD3⁺ FoxP3⁺ T cells were quantified in the whole tumor section using the same software and expressed as a number of cells/mm² (number of total cells divided by the surface area in mm²). The region of TLS was determined manually by referring to the double staining DC-Lamp/CD3 to determine the TLS area. The DC-Lamp/CD3 staining was performed on the serial tumor sections from the same set of patients. The surface area of the TLS was also determined by the Calopix software. The density of CD3⁺ FoxP3⁺ cells in TLS was determined with an automatic counting using Calopix” (page 2 line 20 - page 3 line 5).

The text was modified accordingly, as followed “... we aimed to determine the prognostic importance of TIL-Tregs (i.e. total Tregs on the whole tumor section) according to these two variables” (page 19, lines 377-378). The figure legend 8 was changed, too as “Densities of total of CD3⁺ FoxP3⁺ Tregs (whole tumor section i.e. Tregs infiltrating all sub-areas of the tumor), TLS-DC-Lamp⁺ mature DC, and CD8⁺ T cells were determined using serial sections of tissues (n=338 NSCLC)” (page 35, lines 794-796).

7. Supplementary Figure 5. These graphs are essential for the conclusions of the manuscript (and are even mentioned in the title!) and should be presented as a main figure instead (Figure 9). An additional survival graph with only TLS-DC/CD8hi, TLS-DC/CD8mix, and TLS-DC/CD8lo (without Treg data) should be shown in the same figure for comparison.

MCD: We fully agree with the remark of the Reviewer. The Supplementary Figure 5 is now the Figure 9 (and Supplementary Figure 6 is now therefore Supplementary Figure 5) in the revised manuscript, as requested. New Kaplan-Meier curves for TLS-DC + CD8 have been added (new Fig. 9f, see below).

These curves show that the median survival of the « best » group of patients (TLS-DC^{high} CD8^{high}, black curve) is 92 months whereas it is not reached for the combination with total Tregs (blue curve, Fig. 8f) or with TLS-Tregs (blue curve, Fig. 9d). All together, these data demonstrate that the combination of the three parameters, Treg, TLS-DC and CD8, provides a strong prognostic indicator of survival.

Reviewer #2 (Remarks to the Author):

I am impressed by authors efforts to clarify and rebut many significant points raised in a review of their paper. The manuscript has clearly improved in quality, it is much easier to recognize and appreciate the novelty brought by the study.

The authors have provided significant evidence of the specificity of the CD62L staining as a TLS marker, in the form of newly established multiplex IF staining. This strengthened validity of major findings from the transcriptome analysis. The potential for re-invigoration of the Tregs (associated with the TLS) by anti-CTLA4 and anti-GITR antibodies, but not other immune cell receptors, is an important piece of information in light of clinical efforts in the field of cancer immunotherapy. I approve the manuscript for the publication in the current version.

Reviewer #3 (Remarks to the Author):

MCD: First, we would like to thank Reviewer 3 for his/her comments and suggestions to improve the quality of our manuscript. Please find below our answers to the first and second rounds of questions.

Devi-Marulkar et al. show that Tregs have similar immune profiles in TLS (lung tumors) and non-TLS (distant normal tissue) areas of the tissues examined. They further show that the Treg TIL are functional and their suppressive abilities can be abrogated by blocking the immune checkpoint molecules CTLA-4 and GITR. They find that Tregs in the tumor, but particularly in the TLS, are associated with poor clinical outcome. And TLS Tregs together with mature DC or CD8 T cells distinguish lung cancer patients with the highest risk of death.

Comments on the original manuscript:

The data presented in this manuscript are interesting but due to their primarily descriptive nature they do not greatly advance our understanding of the interplay between Treg and conventional T effector cells in the tumor microenvironment and TLS. The group of tumors analyzed are extremely heterogeneous in the balance between Treg and conventional CD4 T cells. The authors could have reinforced their

conclusions by taking these descriptive observations one step further to link the balance of Treg to Tconv in a specific region (ex TLS or non-TLS, B cell follicle or T cell zone of a TLS, etc) and between tumors with more/less TIL and or TLS and survival.

MCD: We would like to make a more general comment before our point-by-point response to the specific comments of the Referee #3. Our study is not a study performed using murine models. It is performed with biopsies and blood obtained from NSCLC patients, a study that is strictly regulated in terms of what can be done with the samples, the types of samples that can be obtained (size, tissues...). Thus, it has severe limitations in terms of experiments that can be performed with the collected samples, in particular with regard to functional studies or to the use of samples from other tissues. Anyhow, we tried our best to answer the queries of the Reviewer by introducing the results of new experiments and analyses in the manuscript.

This study examines many facets of TLS in the tumor but does not provide insight into what is happening in the balance between conventional T cell activation and Treg-mediated suppression that leads to better or worse survival.

MCD: The balance between the activation of conventional T cells vs the Treg-mediated suppression is discussed in details below (see answers to the comments 3 and 6).

Specific questions:

1. The similarities between Tregs in tumors and non-tumor distant lung suggests that these tissue localized cells have differentiated and matured similarly before entering the tissue. What about other tissues outside of the lung – are they similar or different? Do they also show this activated, effector-memory phenotype? Is this just a reflection of their ability to enter tissue locations and as such is not specifically influenced by the tumor microenvironment?

MCD: We would like be able to answer these questions. However, in these human studies, it was not authorized to collect tissues other than blood and lymph nodes of the enrolled NSCLC patients for obvious ethical and clinical reasons.

2. Is there a difference in activity between TLS with higher levels of Tregs compared to those with lower levels? In other words, are these TLS less active in other ways?

MCD: To our knowledge, only the data published by Joshi and his colleagues (Joshi et al., Immunity, 2015) partially answered to that issue. Using a murine model of lung cancer, these authors demonstrated that some Tregs are located in tumor-associated TLS of the lungs, and that the depletion of Tregs in the tumor is associated with a strong anti-tumor activity. The authors of this study hypothesized that Tregs present in the TLS are important players in the regulation of local anti-tumor responses. This work is reported in the discussion section as followed “A preclinical mouse model

with lung adenocarcinoma demonstrated the immunosuppressive role of the Tregs in lung tumor-associated TLS (46). This study showed that Treg depletion, in the lung tumor-bearing mice, improves the anti-tumor response and infiltration of tumor antigen-specific T cells in the tumor and induces the destruction of the tumors *via* a protective response generated in the TLS” (page 24, lines 494-498, ref. 46).

3. Is the sorted population of Tregs variable in the number of suppressive relative to activated T cells? Did you check Foxp3 expression (protein by flow cytometry) and gene status (demethylation)? Differences in the ratio of activated to suppressive CD2⁺CD4⁺CD8⁻CD25^{hi}CD127⁻ T cells could alter the activity of the TLS.

MCD: T cells infiltrating tumors do not have an activated or an immunosuppressive phenotype but express both so-called inhibitory or activating receptors at variable levels. Thus, it is hazardous to conclude about suppression versus activation only based on the number of suppressive vs activated T cells.

Regarding Treg sorting, we isolated them using a combination of cell surface markers (CD2⁺ CD4⁺ CD8⁻CD25^{hi} CD127⁻), and their purity was checked after each sorting. This workflow was developed after assessing that these cells express high level of FoxP3 protein by flow cytometry, as shown in Figure 2b (middle panel). We would like to underline that the absolute number of sorted Tregs is very low in most tumors, thus preventing to perform demethylation analysis in addition to the gene expression analysis. Up-coming technologies such as scRNAseq coupled to protein detection (ex. “CITE-seq” for cellular indexing of transcriptomes and epitopes by sequencing and “REAP-seq” for RNA expression and protein sequencing assay) will certainly help answering this question.

4. What exactly are the areas defined as non-TLS? Are they aggregates where the CD4, CD8 and B cells are randomly arranged or are they areas where there are congregations of T cells or B cells that could be considered small groups?

MCD: TLS are structurally organized entities, mainly composed of T and B cells with a T cell-rich zone adjacent to a B cell-rich zone in the tumor stroma (cell-cell contact). B cell follicle also contains a network of follicular dendritic cells, as previously published (Dieu-Nosjean et al., J Clin Oncol, 2008). In addition, NK cells have never been observed in TLS. Non-TLS areas is considered as being the remaining stroma once TLS areas has been excluded. The immune infiltrate of the non-TLS areas is highly heterogeneous between patients harboring tumors of the same histological type, and can contain scattered CD4⁺ and CD8⁺ T cells, memory B cells, plasma cells, macrophages, NK cells, and neutrophils randomly located in the stroma. Importantly, follicular dendritic cells are never detected in the non-TLS areas. Some heterogeneous clustering of these immune cell populations can be observed, but they are never organized and segregated, as observed in TLS (see Fig. 1g, lower panel).

5. In the TLS, are the Tregs more localized in the B cell follicle or in the T cell zone?

MCD: Tregs were observed in both T- and B-cell zones of the TLS, and more frequently in the latter zones.

6. Is the heterogeneity of Tregs relative to conventional T cells a reflection of their overall ability to suppress the activities of the latter? Or are there other factors in the

tumor that contribute to immunosuppression? It is interesting that there is similar heterogeneity in the LN although the ratios are lower in the LN. Can you comment on their known activities in LN's?

MCD: Figure 3a shows that the percentage of Tregs/total CD4⁺ T cells is higher in tumors than in LN indicating that tumors are more infiltrated by Tregs compared to LN (among the CD4⁺ T cell compartment). Figures 3c and 5d show that the stage of differentiation and activation of Tregs in the tumors is heterogeneous indicating that these cells will not exhibit the same immunosuppressive ability. We also show that TIL-Tregs migrate into distinct areas of the tumor mass, i.e., tumor nests and tumor stroma including TLS. Our present data obtained from patients (including data about survival) and one murine study (Joshi et al., *Immunity*, 2015) suggest that Tregs can inhibit the immune responses taking place in NSCLC-associated TLS. In addition, Tregs also home in non-TLS areas where they can inhibit the anti-tumor activity of effector cells such as CD8⁺ T cells. The tumor microenvironment is a milieu where the equilibrium between cells with regulatory and effector functions is tightly regulated over the time. In the LN, Tregs can exert their immunosuppressive activity against several immune cells such as mature DC, T and B cells by controlling inflammatory response and preventing autoimmunity. Thus, although a number of characteristics about the heterogeneity of Tregs relative to conventional T cells are shared by TLS and LN, the tumor microenvironment exhibits a very complex immune contexture, marked by a strong cell plasticity and mobility within the tissue, where different players, among which Tregs, are capable of exerting a strong immune suppression of anti-tumor responses.

7. Is there any evidence that the Treg are differentiated from conventional T cells in the tissues compared to being recruited after differentiation?

MCD: There is no data in the literature demonstrating whether TLS-Tregs extravasate from the blood through HEV, or whether they differentiate from naïve T cells selectively present in the T-cell areas of the TLS. Further experiments such as spectral transcriptomic analysis coupled to TCR repertoire analysis on tumor section will give the answer but, which represents costly experiments to be performed on a significant number of patients.

8. Are the differences in gene expression comparing Tregs from blood with tumors relevant? The cells in the blood are known to not be active. Because there were less pronounced differences in the comparison with LN and distant normal tissues, maybe the differences with the blood are just alterations associated with infiltrating tissues in general and not specific to the tumor?

MCD: We compared the transcriptomic signature of Tregs isolated from tumor, non-tumor distant lung, draining lymph node (LN) and blood of NSCLC patients. We observed that the transcriptomic signature and the phenotype of Tregs are similar between Tregs isolated from non-tumor distant lung and tumor, indicating that they may exert similar functions. This observation is in line with pathologists who argue that non-tumor distant lung cannot be considered as a normal tissue due to the observed tissue remodeling, a consequence of the sensing of the tumor-induced inflammation by local immune and non-immune cells.

The comparison between tumors and draining LN shows that the level of expression of most genes in sorted Tregs is not statistically different. This observation is in

agreement with the fact that both the tumor and the relevant draining LN are known to be critical for the (re)activation and proliferation of tumor-specific T cells. However, anti-tumor T cells represent only a minor percentage of total peripheral blood T cells, even after vaccination. Because blood Tregs are mainly resting, it can explain the high differences in gene expression when comparing blood Tregs versus TIL-Tregs. As stated above, it was not permitted due to clinical and ethical consideration, to get other tissues (unrelated to tumor) from NSCLC patients to demonstrate that the transcriptomic signature of Tregs isolated from tissues does not reflect solely alterations associated with the infiltration of tissues.

9. The molecular profiles of Tregs (compared to conventional CD4 T cells) in and outside of the TLS are interesting but highly descriptive. It would be nice to see some functional link between expression data and activity. Are there any differences in surface receptors (like CXCR3 or others that differ in staining intensity) that could be used to sort and test them separately in the functional assay?

MCD: CD62L is the best marker to discriminate Tregs in *versus* outside TLS as shown in Figure 1 (protein expression) of the manuscript. Indeed, this cell surface marker shows the highest P-value as well as the highest fold change among all markers tested in our study (Fig. 5c, mRNA expression).

We wanted to perform co-cultures between conventional T cells and TLS-Tregs - or non-TLS Tregs - in order to see some functional link transcriptomic signature and Treg activity according to their *in situ* location but the number of TLS-Tregs that could be purified from tumor biopsies was too low to perform ex vivo functional assays (e.g. T cell proliferation, cytokine secretion, production of cytolytic granules).

10. What does it mean that only blocking GITR or CTLA-4 but not ICOS, PD-1, TIGIT or TIM-3 reversed Treg-mediated suppression of conventional T cells. If suppression is TGF β or IL-10 mediated then is there a link?

MCD: We have tested whether anti-ICP neutralizing antibodies could restore the proliferative capacity of conventional T (convT) cells in co-culture with Tregs based on the strong expression of the corresponding ICP on TIL-Tregs that we observed. Antibodies were tested separately or in combination. Only anti-GITR or anti-CTLA-4 antibodies neutralized the Treg suppressive activity. We can exclude a direct effect of the ICP neutralizing antibodies on convT cells as no impact of these antibodies on convT cells cultured alone was observed.

We cannot exclude that IL-10 may exert an immunosuppressive effect, although the concentration of this cytokine in the supernatants of tumor-infiltrating Treg-Tconv co-cultures (n=5 NSCLC) was extremely low (< 6 pg/mL) in all conditions, as determined by Cytometric Bead Array (CBA).

The human Th1/Th2/Th17 cytokine kit (commercial kit) does not include TGF- β , and we have no data about its concentration which is known to be hard to determine due to its weak stability. However, we performed new flow cytometry analysis on fresh NSCLC samples. The figure below shows that only one third of TIL-Tregs are positive for TGF- β , and most of Tregs TGF- β ⁺ express CTLA-4, ICOS and PD-1 which may not explain functional differences observed with some blocking antibodies.

Further investigation will be required in order to decipher the mechanism by which only GITR or CTLA-4 but not PD-1, ICOS, TIGIT, or Tim-3 blockade impacts Treg biology. The heterogeneity of the suppressive effects of these various ICP and their relative role in a number of clinical situations is still a matter of controversy. This issue could be investigated using tumor-bearing mice deficient for each of this ICP.

11. What is the difference in a TLS with low Tregs, high DC and high CD8 compared to a high Treg, low DC, low CD8 TLS? Are there other characteristics that vary between these two types of TLS that could also reflect survival?

MCD: The work published by Joshi *et al.* (Immunity, 2015) shows that Tregs in tumor-associated TLS actively suppress immune responses. This setting is likely to correspond to that of NSCLC patients exhibiting TLS containing high Tregs, low mature DC and low CD8⁺ T cell infiltrates. Following Treg depletion, these authors observed an increase in TLS number and size as well as a strong T cell infiltration. This experiment mimics the situation in which NSCLC patients exhibit TLS with low Tregs, high mature DC and high CD8⁺ T cell infiltrate. Overall, Joshi *et al.* demonstrated that a local Treg depletion is associated with anti-tumor responses and correlates with a better survival. More recently, we reported that high TLS-B cells density is associated with lower Treg frequency, and combination of TLS-B cells and Treg densities was found to be a strong prognostic indicator of the clinical outcome of NSCLC patients (Germain *et al.*, Frontiers Immunol, 2021). All together, we believe that Treg in TLS can exert a negative impact on both cellular and humoral immune responses, with important consequences on patient survival. Immunotherapy targeting Tregs may boost ongoing endogenous anti-tumor immunity in TLS.

12. The speculation in the discussion should be removed.

MCD: This query is the same as in comments 5 and 6 below. See our answers below.

Comments on the revised manuscript:

While I found the manuscript greatly improved in response to Reviewer 1 & 2's comments there are still some open questions.

1. The authors argue that the presence of regulatory T cells (high versus low) in the TLS is associated with a poor prognosis. As an extension of the comment on Treg/Tconv balance mentioned in the original review comments (underlined), this subject has not been addressed in the revised manuscript. There are multiple studies of viral infection, autoimmune disease and more recently cancer, showing that the balance of T effector cells to T regulatory cells is an important determinant of activation or suppression. How do the authors explain this? Can they calculate the balance between Treg and Tconv in their samples and correlate this with response? This would be extremely informative.

MCD: We thank the Reviewer for this suggestion. We established the prognostic value of total CD8⁺ T cells-to-total Treg ratio as well as total CD8⁺ T cells-to-TLS Treg ratio in this cohort of NSCLC patients (new Kaplan-Meier curves). For each analysis i.e. considering total Treg or only TLS-Treg, we observed that NSCLC patients having a high ratio have the best clinical outcome. These data are in accordance with the literature, and again indicate that the balance of effector T cells to regulatory T cells is critical for the control of the tumor growth.

These new data are included in the present revised version of the manuscript (page 20 lines 401-404, Fig. 8g and 9e) as followed: “Similarly, high CD8⁺ T cells-to-Tregs ratio (Fig. 8g) or high CD8⁺ T cells-to-TLS Tregs ratio (Fig. 9e) correlated with long-term survival of patients indicating that the balance of effector T cells to regulatory T cells is critical for the clinical outcome of patients“. Figures 8 and 9 are fully dedicated to Kaplan-Meier analysis with total Tregs and TLS-Tregs, respectively.

2. The authors used a customized array for the gene expression study and limited markers in flow cytometric analysis. It is unfortunate that they did not include other important markers of TLS such as CXCR5 [this could have been used in flow in combination with CD62L to show whether the TLS-associated Treg were in fact all or partially Tfr (T follicular regulatory cells)]. Multiple studies have shown that TLS contain not only CXCR5⁺ B cells but also CXCR5⁺ T cells, both CD4 and CD8.

MCD: Please find below new flow cytometry analysis showing that TLS-Tregs express lymphoid chemokine receptors (CCR7, CXCR5) as well as Tfh markers such as PD-1 and ICOS. Data are included to the new version of the manuscript as followed: “We confirmed that TLS-Tregs also express lymphoid chemokine receptors (CCR7, CXCR5) as well as Tfh markers such as PD-1 and ICOS (Fig. 2d)“ (page 13, lines 258-259, and Fig. 2d).

3. It is curious that the authors find Tcomv are mostly naïve and central memory T

cells in TLS – many studies show that they are more mature (EM1-EM4) memory cells. Can they comment on this?

MCD: This comment refers to Fig. 3c (lower panel) where the highest percentages of conventional CD4⁺ T cells with naïve and central-memory (CM) phenotypes are detected in TLS. As observed in canonical lymphoid organs, TLS are a key site for the activation of naïve T cells as well as for the reactivation of central-memory T cells. Actually, the data reported in the present manuscript are in agreement with the literature (de Chaisemartin et al., *Cancer Res*, 2011; Kleinwort et al., *IOVS*, 2016). Other authors also reached the same conclusions as we do herein:

- Cabrita et al., *Nature*, 2020: “In single-cell data, B-cell-rich samples contained more CD4⁺ and CD8⁺ T cells with naïve and/or memory-like characteristics as compared to B-cell-poor samples, suggesting an influx of naïve and memory T cells to TLS” in melanoma patients.

- Sautes-Fridman et al., *Nature Rev Cancer*, 2019: “... naïve T cells, however, were enriched in the TLS when compared with the rest of the tumour”.

- Engelhard et al., *J Immunol*, 2018: “tumor-associated TLS are sites for generating useful antitumor immune responses in newly entering naïve T and B cells”.

- Peske et al., *Nature Comm*, 2015: “these results suggest that organized lymphoid tissue (i.e. TLS) associated with LN-like vasculature (i.e. HEV) may serve as sites for the activation and regulation of recently entering naïve T-cells” in B16-OVA and LLC-OVA models.

Finally, the upper panel of Figure 3c shows that conventional CD4⁺ T cells in non-TLS areas mostly exhibit an effector-memory phenotype (EM1, EM4 and to a lesser extent EM3) in accordance with the presence of effector cells close to tumor cells.

4. Is the higher expression of PD-1 on conventional T cells compared to Treg due to a greater follicular helper T cell presence?

MCD: There is no correlation between high PD-1 expression on conventional T cells and Tfh presence by flow cytometry.

5. Discussion p22, lines 425-8: There is no demonstration that CCR4 and the ligands CCL17 and CCL22 are expressed as protein in Tregs so this statement is speculation without even a reference to back it up (FACS, IF, or the chemokines in the SN).

MCD: this sentence was removed from the discussion (page 22, line 446).

6. Discussion p24, lines 485-8 is again speculation based on gene expression analysis and needs a demonstration that CXCR3 protein is expressed on the surface of these cells.

MCD: We thank the Reviewer 3 for this comment. You will see below new data performed on NSCLC samples, and showing the expression of CXCR3 on Th1 T cells (CD3⁺ CD4⁺ CD8⁻ CD127⁺ T-bet⁺, left panel) and Tregs (CD3⁺ CD4⁺ CD8⁻ CD127⁻ FoxP3⁺⁺ CD25⁺⁺, right panel) by flow cytometry. A dim expression of CXCR3 is observed on both T cell subsets whereas a bright expression is only detected on Th1 T cells. Thus, Th1 and Tregs positive for CXCR3 have the capacity to migrate in response to CXCL9, CXCL10 and CXCL11 in the tumor microenvironment. This data is included in the new version as followed “As for Th1 T cells, most Tregs express CXCR3 protein but only at an intermediate level (Fig. 5e).” (pages 17-18, lines 342-343, Fig. 5e), and the initial sentence in the discussion is maintained, accordingly.

REVIEWERS' COMMENTS:

Reviewer #1 (Remarks to the Author):

The manuscript has been further improved. All my concerns were satisfactorily addressed by the authors.

Reviewer #3 (Remarks to the Author):

The authors have persuasively answered the questions and remarks from my previous review comments. It is clear that patient samples have restrictions not posed by studies using murine models, and while this reviewer well understands those constraints the reviewer still considers that functional studies using human material, while limited, are feasible with some new and innovative small scale technical approaches. Efforts to implement them are important, but this is not a decisive issue for this manuscript. The clarity of the data and the arguments and interpretations in this revised version make the current manuscript acceptable for publication.